Resource

# Quantitative assessment of the nanoanatomy of the contractile vacuole complex in *Trypanosoma cruzi*

Ingrid Augusto[1],*, Wendell Girard-Dias[1,3],*, Alejandra Schoijet[5,6], Guillermo Daniel Alonso[5,6], Rodrigo V Portugal[7,8], Wanderley de Souza[1,2,4], Veronica Jimenez[9], Kildare Miranda[1,2,4]

***Trypanosoma cruzi* uses various mechanisms to cope with osmotic fluctuations during infection, including the remodeling of organelles such as the contractile vacuole complex (CVC). Little is known about the morphological changes of the CVC during pulsation cycles occurring upon osmotic stress. Here, we investigated the structure–function relationship between the CVC and the flagellar pocket domain where fluid discharge takes place—the adhesion plaque—during the CVC pulsation cycle. Using TcrPDEC2 and TcVps34 overexpressing mutants, known to have low and high efficiency for osmotic responses, we described a structural phenotype for the CVC that matches their corresponding physiological responses. Quantitative tomography provided data on the volume of the CVC and spongiome connections. Changes in the adhesion plaque during the pulsation cycle were also quantified and a dense filamentous network was observed. Together, the results suggest that the adhesion plaque mediates fluid discharge from the central vacuole, revealing new aspects of the osmoregulatory system in *T. cruzi*.**

## Introduction

Cell volume control is crucial for maintaining overall cell homeostasis, governing different aspects of cellular behavior and function. Such regulation is intertwined with various processes, including alterations in ion concentrations, distribution of osmolytes, proteins and peptides, and activation of signaling pathways (Suescún-Bolívar & Thomé, 2015). These changes can also induce cellular responses that may modulate the formation of biomolecular condensates (phase separation) and membrane distension, and lead to the activation of mechanosensitive receptors (Sehgal et al, 2022; Morishita et al, 2023). Such processes are essential for the ability of different cells, including protozoan parasites, to respond to and integrate cues from the environment, thereby maintaining volume homeostasis and securing their division cycle both in vivo and in vitro (Wang et al, 2020).

*Trypanosoma cruzi* is a pathogenic protozoan causative agent of Chagas disease, listed by the World Health Organization among Neglected Tropical Diseases. Chagas disease impacts ~7 million people globally, with 75 million at risk, and results in about 12,000 deaths annually, according to the World Health Organization. The current chemotherapy against this parasite, which includes the use of benznidazole and nifurtimox, faces significant challenges such as low specificity, high toxicity, and drug resistance (Field et al, 2017). Understanding the basic cell biology of this parasite is crucial for pinpointing potential biochemical targets that could lead to the development of safer and more efficient therapies. This is particularly significant when it comes to mechanisms related to cell survival, such as osmoregulatory control. Furthermore, because *T. cruzi* is an early divergent eukaryotic organism, it serves as an excellent model for investigating essential cellular processes and how they have evolved in higher eukaryotes.

Along its developmental cycle, *T. cruzi* assumes three primary forms, which adapt to different hosts and environments. At each stage, the parasites encounter varying levels of osmotic stress. Metacyclic trypomastigotes, for instance, exit the gut of the insect vector in an extremely hyperosmotic environment (measuring up to 1,000 mOsm/kg in some cases) and, upon infection, quickly reach the bloodstream of the vertebrate host, being exposed to environments with lower osmotic pressure (around 300 mOsm/kg) (Kollien et al, 2001). Variations in osmotic pressure also occur when the blood is passing through different organs (i.e., kidney) and upon invasion of host tissues. In the invertebrate host, fluctuations

[1]Laboratório de Ultraestrutura Celular Hertha Meyer, Centro de Pesquisa em Medicina de Precisão, Instituto de Biofísica Carlos Chagas Filho and Centro Nacional de Biologia Estrutural e Bioimagem, Universidade Federal do Rio de Janeiro, Rio de Janeiro, Brazil [2]Instituto Nacional de Ciência e Tecnologia em Biologia Estrutural e Bioimagem – Universidade Federal do Rio de Janeiro, Rio de Janeiro, Brazil [3]Plataforma de Microscopia Eletrônica Rudolf Barth, Instituto Oswaldo Cruz – Fiocruz, Rio de Janeiro, Brazil [4]Centro Multiusuário para Análise de Fenômenos Biomédicos, Universidade do Estado do Amazonas, Manaus, Brazil [5]Instituto de Investigaciones en Ingeniería Genética y Biología Molecular "Dr. Héctor N. Torres", Buenos Aires, Argentina [6]Departamento de Fisiología, Biología Molecular y Celular, Facultad de Ciencias Exactas y Naturales, Universidad de Buenos Aires, Buenos Aires, Argentina [7]Laboratório Nacional de Nanotecnologia, Centro Nacional de Pesquisa em Energia e Materiais, Campinas, Brazil [8]Programa de Biotecnologia, Universidade Federal do ABC, Santo André, Brazil [9]Department of Biological Sciences, College of Natural Sciences and Mathematics, California State University Fullerton, Fullerton, CA, USA

Correspondence: kmiranda@biof.ufrj.br
*Ingrid Augusto and Wendell Girard-Dias contributed equally to this work

in the osmotic pressure of the extracellular medium happen in various segments of the digestive tract of the insect vector and specific mechanisms that shield the parasite from these environmental fluctuations have been identified (Kollien et al, 2001; Jimenez, 2014), as detailed below.

In different trypanosomatids, the initial response to osmotic stress involves the rapid release of amino acids and other inorganic osmolytes to the extracellular medium, presumably through membrane channels and transporters (Vieira et al, 1996; Blum et al, 1999; Rohloff et al, 2003; LeFurgey et al, 2005). This mechanism is responsible for ~50% of cell volume recovery in *T. cruzi*. The remaining half involves three structures: (i) acidocalcisomes, organelles rich in ions and polyphosphate chains, which may influence the formation of an osmotic gradient (Rohloff et al, 2004; Rohloff & Docampo, 2008); (ii) the contractile vacuole complex (CVC), which regulates cytoplasmic water content by controlling water uptake into the organelle and efflux via (iii) the flagellar pocket (FP) (Rohloff & Docampo, 2008). The latter comprises a surface domain formed by a plasma membrane invagination, where intense endocytic and secretory activities take place (Landfear & Ignatushchenko, 2001; De Souza, 2002).

In kinetoplastids, the CVC comprises interconnected tubules and vesicles collectively known as the spongiome. These structures are connected to the central vacuole (CV) located in the vicinity of the flagellar pocket (Linder & Staehelin, 1977; Attias et al, 1996; Jimenez et al, 2022). Initial morphological descriptions were primarily based on electron microscopy (EM) protocols that used room temperature chemical fixation, a method known to generate various artifacts because of the slow diffusion rates of the fixatives, and the osmotic effects resulting from the loss of plasma membrane selectivity. Consequently, EM images of the CVC often display significant structural alterations, that is, the collapse of tubules and vesicles of the spongiome. A reliable structural characterization of this organelle was achieved using cryofixation protocols, which confirmed the presence of only one CVC in non-dividing *T. cruzi* cells, formed by an organized network of interconnected tubules linked to the central vacuole and localized in the anterior region of the parasite (Girard-Dias et al, 2012). Previous EM images suggested the existence of an electrodense anchoring domain localized between the vacuole and the flagellar pocket, resembling the adhesion plaque initially described in *L. collosoma* (Linder & Staehelin, 1979; Girard-Dias et al, 2012). Since this first description, little progress has been made in the characterization of the CVC structure and how it translates to its function (Jimenez et al, 2022). Information regarding the fine structural changes that occur in the CVC during its pulsation cycle, spanning from the uptake of intracellular water to its discharge is still lacking. In addition, the structural organization of the adhesion plaque in this parasite has not yet been addressed.

The current model for signaling responses to osmotic challenges in *T. cruzi* suggests that the activation of adenylate cyclase is stimulated by the influx of water after hypoosmotic stress (Rohloff & Docampo, 2008; Chiurillo et al, 2023). Adenylate cyclase interacts with cAMP response protein 3, increasing the intracellular levels of cyclic AMP (cAMP) (Chiurillo et al, 2023). As a result, acidocalcisomes fuse with the central vacuole,

transferring aquaporins and establishing an osmotic gradient (Rohloff et al, 2004; Docampo, 2016; Niyogi & Docampo, 2015; Docampo, 2024). The fluid accumulated in the vacuole is discharged to the flagellar pocket (Rohloff & Docampo, 2008). In this context, mutants overexpressing enzymes that regulate this pathway such as TcrPDEC2 and TcVps34 (TcrPDEC2 OE and TcVps34 OE) serve as valuable tools for exploring the morphofunctional organization of the osmotic regulation system in *T. cruzi* (Schoijet et al, 2019). The phosphodiesterase TcrPDEC2 plays a pivotal role in modulating the concentration of cAMP around the CVC, regulating the activity of aquaporins and the fusion of the spongiome with the central vacuole (Schoijet et al, 2011). TcrPDEC2 contains an FYVE domain that determines its subcellular localization and is essential in maintaining its catalytic activity (Schoijet et al, 2011). This domain binds to membranes rich in phosphatidylinositol 3-phosphate (PI3-P), produced by a phosphatidylinositol 3-kinase (TcVps34), thereby localizing TcrPDEC2 to the CVC, primarily in the spongiome (Schoijet et al, 2011). TcrPDEC2 exerts a negative modulatory effect on the osmotic signaling pathway, and its overexpression results in mutants with diminished efficiency in recovering cell volume after hypoosmotic shock (Schoijet et al, 2011). On the other hand, overexpression of TcVps34 results in superefficient mutants (presumably delocalizing TcrPDEC2 because of the inclusion of PI3-P in another cell membrane) with faster cell volume recovery when subjected to hypoosmotic stress (Schoijet et al, 2008).

Taking advantage of these phenotypes, here, we compared the structure of the adhesion plaque and the primary changes that occur in the CVC during osmoregulation in WT cells and mutants that are less efficient (TcrPDEC2 OE) and superefficient (TcVps34 OE) in osmoregulation. Using high-resolution quantitative 3D electron microscopy, electron tomography, and focused ion-beam (FIB) SEM, we offer a comprehensive analysis of the structure of the CVC-FP domain, providing insights into the basic organization of these two structures involved in the osmoregulatory system in *T. cruzi*. Understanding the cell biology of this parasite may shed light on evolutionary conserved events in more complex organisms.

# Results

## The CVC in *T. cruzi* is a dynamic organelle in constant structural reorganization

In *T. cruzi* epimastigotes under isosmotic conditions, the central vacuole was generally more elliptical (Fig 1A and C), whereas under hypoosmotic stress it was larger and more rounded in most cells (Fig 1B and D). Variations in the dimensions of the central vacuole were observed even under isosmotic conditions. Similarly, in hypoosmotic conditions, the spongiome tubules showed a greater number of interconnections. Moreover, their preferential orientation, previously identified as opposite to the kinetoplast and parallel to the flagellum (Girard-Dias et al, 2012), became even more pronounced after the cells were subjected to hypoosmotic stress (Video 1). Morphometric analysis of cells

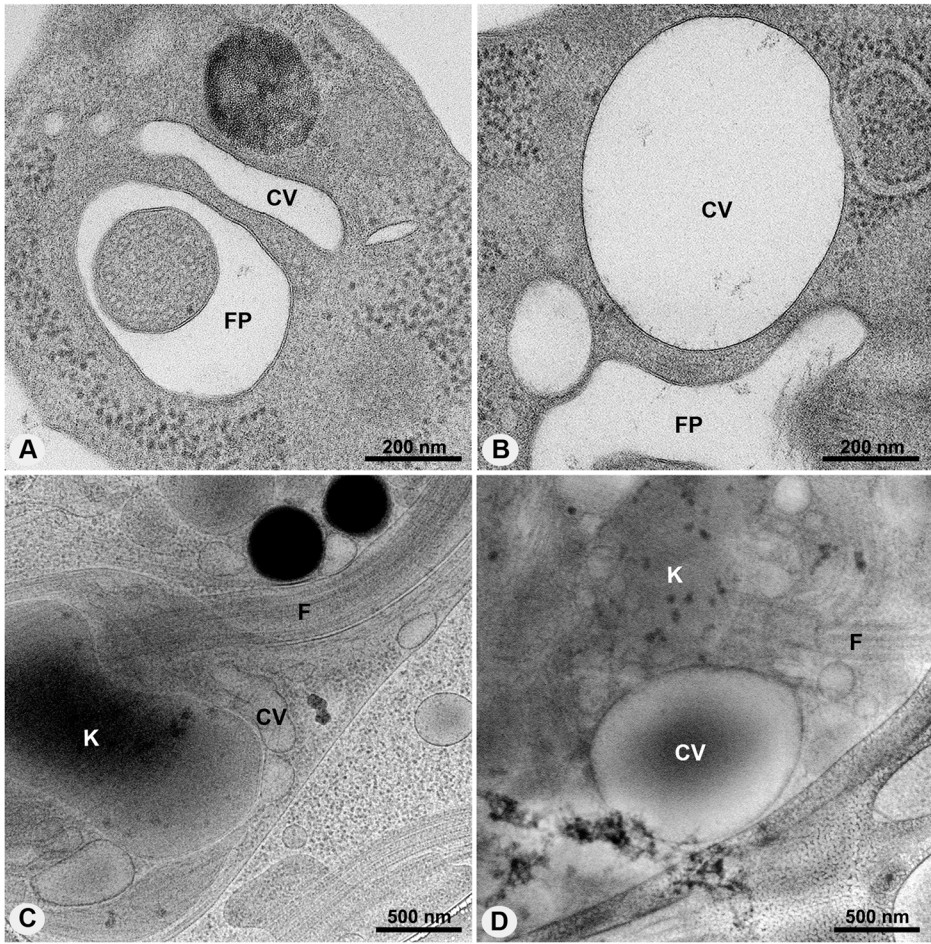

**Figure 1. Organization of the CVC under different osmotic conditions observed in HPF/FS and cryo-EM single images.**
**(A, B)** Ultrathin sections of high-pressure frozen and freeze-substituted cells under isosmotic and hypoosmotic conditions, respectively. Under isosmotic conditions, the central vacuole has an elliptical shape changing to a larger and more rounded shape after hypoosmotic stress. **(C, D)** Cryo-EM single images confirm both conformations of the CVC during osmoregulation. FP, flagellar pocket; F, flagellum; CV, central vacuole; K, kinetoplast.

**Table 1. Morphometric analysis of the central vacuole in epimastigotes of *T. cruzi* under osmotic stress.**

|  | Area (nm² x 10³) | Circularity (nm) | Minimum diameter (nm) | Maximum Diameter (nm) | Average diameter (nm) |
|---|---|---|---|---|---|
| Iso | 77.2 ± 10.6 | 0.68 ± 0[a] | 223.4 ± 18[a] | 460.4 ± 29.8[a] | 341.9 ± 21[a] |
| Hypo | 174.7 ± 17.7 | 0.80 ± 0.04[a] | 381.6 ± 26.2[a] | 591.9 ± 26.1[a] | 486.8 ± 22.1[a] |

The results are expressed as mean ± SEM.
[a]The results that showed significant differences according to *t* test ($P < 0.05$), n = 30.

under both isosmotic and hypoosmotic conditions showed a significant increase in diameter and circularity of the central vacuole after hypoosmotic stress (Table 1).

Morphological variants representative of the different phases of the pulsation cycle were identified, suggesting a presumable temporal sequence of remodeling of the CVC during the regulatory volume decrease (RVD). In phase I, the central vacuole collapsed and exhibited a lamellar shape. At this time point, the spongiome tubules measured between 50 and 100 nm in diameter and showed no connections with the central vacuole (Fig 2A). In phase II, the central vacuole developed a more elongated form and showed increased volume (Fig 2B). Different connections between the spongiome tubules and central vacuole are then visualized (Fig 2B). Spongiome tubules exhibited significant morphological variation, with distended regions ranging in

diameter from 50 to 180 nm. Images suggest active cytoplasmic water absorption by the spongiome and its subsequent transfer to the central vacuole. In phase III, the central vacuole showed higher sphericity and a larger volume than in phase II (Fig 2C). The spongiome tubules displayed a more regular shape, with most tubules measuring ~60 nm in diameter. A few distended tubules with diameters ranging between 60 and 170 nm connected to the central vacuole were, nevertheless, observed. (Fig 2C). In phase IV, the central vacuole presented a circular or oval shape and the largest volume (Fig 2D). The spongiome tubules displayed diameters ranging from 50 to 60 nm with a few connections to the central vacuole (Fig 2D). In the final phase, the CVC initiated fluid discharge, evidenced by a gap of ~25 nm at the adhesion plaque region, when maintaining the integrity of both membranes (Fig 2D), suggesting an initial stage of pore

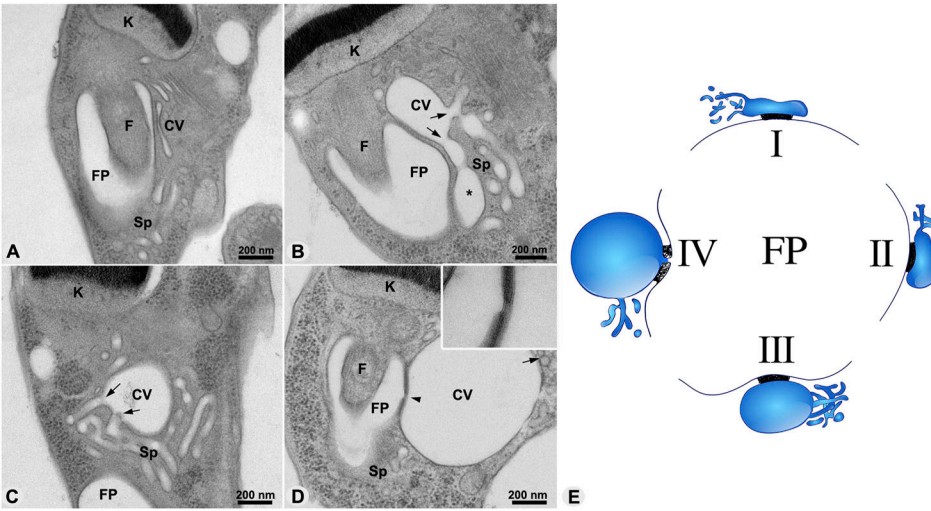

**Figure 2. Remodeling of the CVC structure throughout the pulsation cycle.**
**(A, B, C, D)** Images from ultrathin sections of HPF/FS WT cells. **(A)** Systole phase, characterized by a collapsed central vacuole (CV). **(B)** Initial water influx phase into the CVC, during which the volume of certain spongiome (Sp) tubules (asterisk) increases, with observed connections to the central vacuole (arrows). **(C)** Early diastole phase, showing an increase in the volume of the central vacuole. **(D)** Final diastole phase, where the central vacuole reaches a larger volume. During this phase, water is expelled from the CVC to the flagellar pocket. A region of the adhesion plaque shows a discontinuity in the electrodense area and a decrease in thickness, potentially indicating the formation of a pore (arrowhead and inset). **(E)** Diagram illustrating the four proposed phases transitioning between the systole and diastole stages of the CVC pulsation cycle. In the end stage of systole (I), the central vacuole exhibits a lamellar shape with a few connections to the spongiome, which then increases during the intermediate phase (II) as water begins to flow. The central vacuole then adopts a spherical shape in phase (III), characterizing diastole, until it reaches its maximum volume and diameter in phase (IV), initiating fluid discharge subsequent to the destabilization of the adhesion plaque. FP, flagellar pocket; F, flagellum; CV, central vacuole; Sp, spongiome; K, kinetoplast.

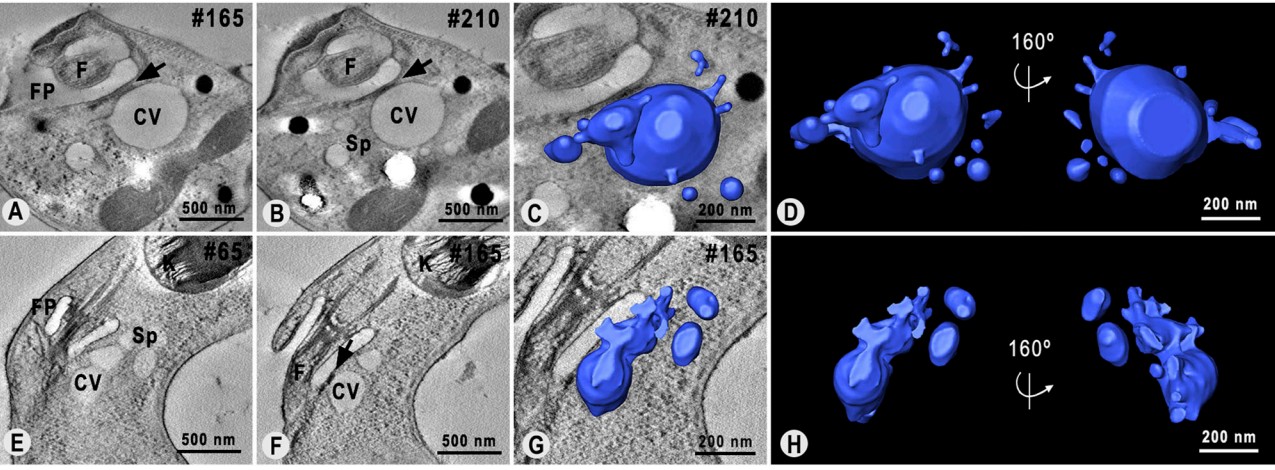

**Figure 3. Serial electron tomography from HPF/FS amastigote and trypomastigote forms.**
**(A, B, C, D, E, F, G, H)** Virtual sections extracted from tomograms and 3D models show the contractile vacuole complex in both amastigote (A, B, C, D) and trypomastigote (E, F, G, H) forms. **(A, F)** Both forms exhibit the central vacuole (CV) in proximity to the flagellar pocket (FP) (A, F), where an electron-dense domain corresponding to the adhesion plaque is observed (arrows). **(C, D)** The 3D model of the amastigote (C, D) reveals the central vacuole in the diastole phase, connected to a spongiome (Sp), featuring interconnections among tubules and vesicles. **(G, H)** In contrast, the central vacuole of the trypomastigote (G and H) exhibits fusion with extended spongiome compartments (asterisk), predominantly characterized by interconnected vesicles. K-kinetoplast, F-flagellum.

formation between the membranes of the CVC and the flagellar pocket. A proposed model for the CVC pulsation cycle (Fig 2E) begins at the end of the systole stage (I), after fluid discharge into the flagellar pocket. The transition period between the end of systole and the start of diastole (II) is marked by the onset of water absorption by the spongiome, which then transfers it to the central vacuole, establishing a few connections. As diastole progresses to phase (III), the central vacuole becomes round and richly connected to the spongiome. At the end of diastole (IV), the central vacuole reaches its maximum volume and circularity but maintains only a few connections to the spongiome, starting fluid discharge.

## The CVC maintains a consistent organizational structure across various developmental stages of *T. cruzi*

High-pressure freezing (HPF) and freeze substitution were also used for the observation of the CVC in other developmental forms of the parasite. The organization and positioning of the CVC in amastigotes feature a central vacuole adjacent to the flagellar pocket, with the spongiome predominantly condensing in the same area, as observed in epimastigotes (Fig 3A–D). In the trypomastigotes, the central vacuole remains close to the flagellar pocket, but vesicles are more prevalent than tubules in the spongiome and the central vacuole is also fragmented (Fig 3E–H).

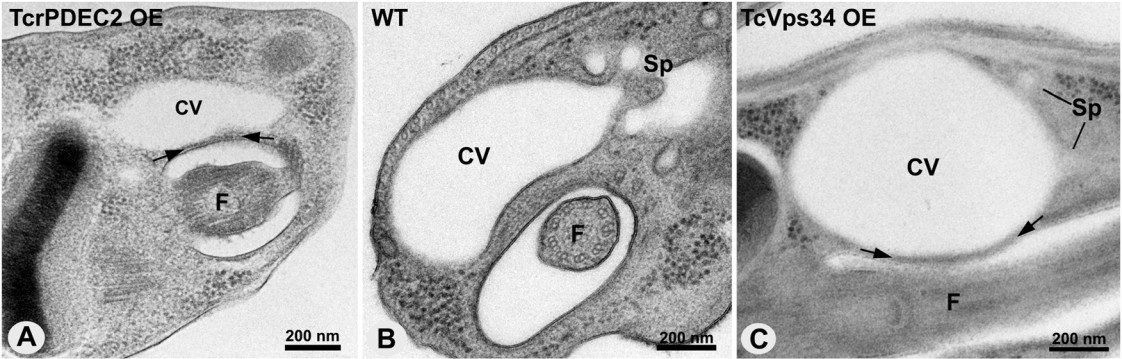

**Figure 4. CVC structures observed in ultrathin sections of HPF/FS WT, TcrPDEC2 OE, and TcVps34 OE mutants.**
**(A)** TcrPDEC2 OE parasite, the spongiome is not visible. **(B)** The WT parasite displays the characteristic organization of the CVC, with vesicles from the spongiome connected to the central vacuole. **(C)** The CVC of a TcVps34 OE cell is depicted with an enlarged central vacuole that has connections to the spongiome. Images are representative of multiple analysis of thin sections, transmission electron tomography and FIB-SEM tomography. CV, central vacuole; F, flagellum; Sp, spongiome.

The adhesion plaque seemed to be a consistent feature across the different developmental stages of *T. cruzi* (Fig 3A, B, and F). These results show that the CVC exhibits a similar structure in both the replicative and infective forms of *T. cruzi*, likely playing an active role in the adaptation mechanisms that respond to shifts in the physicochemical composition of the external milieu along the infection cycle.

### The TcrPDEC2 OE and TcVps34 OE mutants display ultrastructural characteristics that reveal their physiological responses to osmotic stress

To examine the connection between the CVC structure and function in *T. cruzi*, we used two mutants with known differences in their physiological responses to osmotic stress: (i) a mutant over-expressing a phosphatidylinositol 3-kinase (TcVps34 OE), known to have a faster RVD (Schoijet et al, 2008), and (ii) a mutant over-expressing a phosphodiesterase C2 (TcrPDEC2 OE), that presents a slower RVD response (Schoijet et al, 2011). Western blot analysis confirmed that both mutants exhibit significantly higher levels of the overexpressed proteins compared with the WT cells (Fig S1A and B). To better understand the physiological differences in osmotic regulation, we assessed the structural phenotype of the CVC in WT, TcrPDEC2 OE, and TcVps34 OE cells at the same stage of the pulsation cycle (beginning of diastole) (Fig 4). Ultrathin sections confirmed that in both mutants, the central vacuole was consistently situated adjacent to the flagellar pocket, as seen in WT parasites (Fig 4). The adhesion plaque was observed in both mutants (Fig 4A and C). The size of the central vacuole in TcrPDEC2 OE was slightly smaller when compared with WT cells (Fig 4A and B), whereas TcVps34 OE consistently exhibits enlarged vacuoles (Fig 4C).

The three-dimensional organization of the CVC during the systole and diastole stages was quantitatively characterized across all genotypes using electron tomography. Because of the rapid nature of the systole, identifying cells in phase I of the pulsation cycle proved challenging. To facilitate structural analysis of the CVC at the start and end of the pulsation cycle, we categorized both lamellar and elongated central vacuoles as characteristic of the systole stage and all spherical central vacuoles as indicative of the diastole stage. Beyond morphological and spatial analysis, we quantified the continuous area to assess the size and number of fragments composing the CVC, the frequency of connections within its structure, and the volume of both the central vacuole and spongiome during different stages of the pulsation cycle.

The expected shape of the CVC was rarely observed in serial electron tomography reconstructions of TcrPDEC2 OE mutants. At the systole stage, smaller central vacuoles were visualized (Fig 5A), surrounded by a fragmented spongiome formed by small tubules and vesicles (Fig 5A). Morphometric analysis of the continuous area of the CVC revealed a significant number of small fragments ($\leq 0.5 \times 10^5$ nm$^2$) in TcrPDEC2 OE parasites, surpassing what was usually observed in other genotypes (Table 2). On the other hand, in WT cells most of the CVCs in the intermediate systole stage (phase II) presented an elongated central vacuole connected to a spongiome (Fig 5C), preferentially concentrated in the vicinity of the flagellar pocket. In TcVps34 OE mutants, the central vacuole collapsed (Fig 5E), as observed in WT cells. However, in these cells, the central vacuole was larger and the spongiome exhibited tubules that were more widely spread, lacking a polarized organization, and featuring numerous interconnections with the central vacuole (Fig 5E). Compared with WT cells, TcVps34 OE mutants also presented larger fragments, with surface areas ranging between $0.5 \times 10^5$ and $10 \times 10^5$ nm$^2$. These were also the only mutants to have fragment sizes exceeding $20 \times 10^5$ nm$^2$ at the systole stage (Table 2). During the diastole stage (phase IV), TcrPDEC2 OE parasites exhibited a spherical central vacuole with limited connections to the spongiome, which featured some tubules and vesicles in various regions (Fig 5B) when compared with WT (Fig 5D). The limited connectivity was further evidenced by the high number of measured fragments (Table 3). In contrast, TcVps34 OE cells exhibited a significantly larger and rounder central vacuole (Fig 5F) with numerous connections to the spongiome, where tubules were arranged as in the systole stage (Fig 5E and F). In addition, some TcVps34 OE cells were observed to have more than one vacuole connected to the spongiome (Fig 5F).

When comparing the percentage of smaller fragments ($\leq 50 \times 10^3$ nm$^2$) between the systole and diastole stages, there was a slight

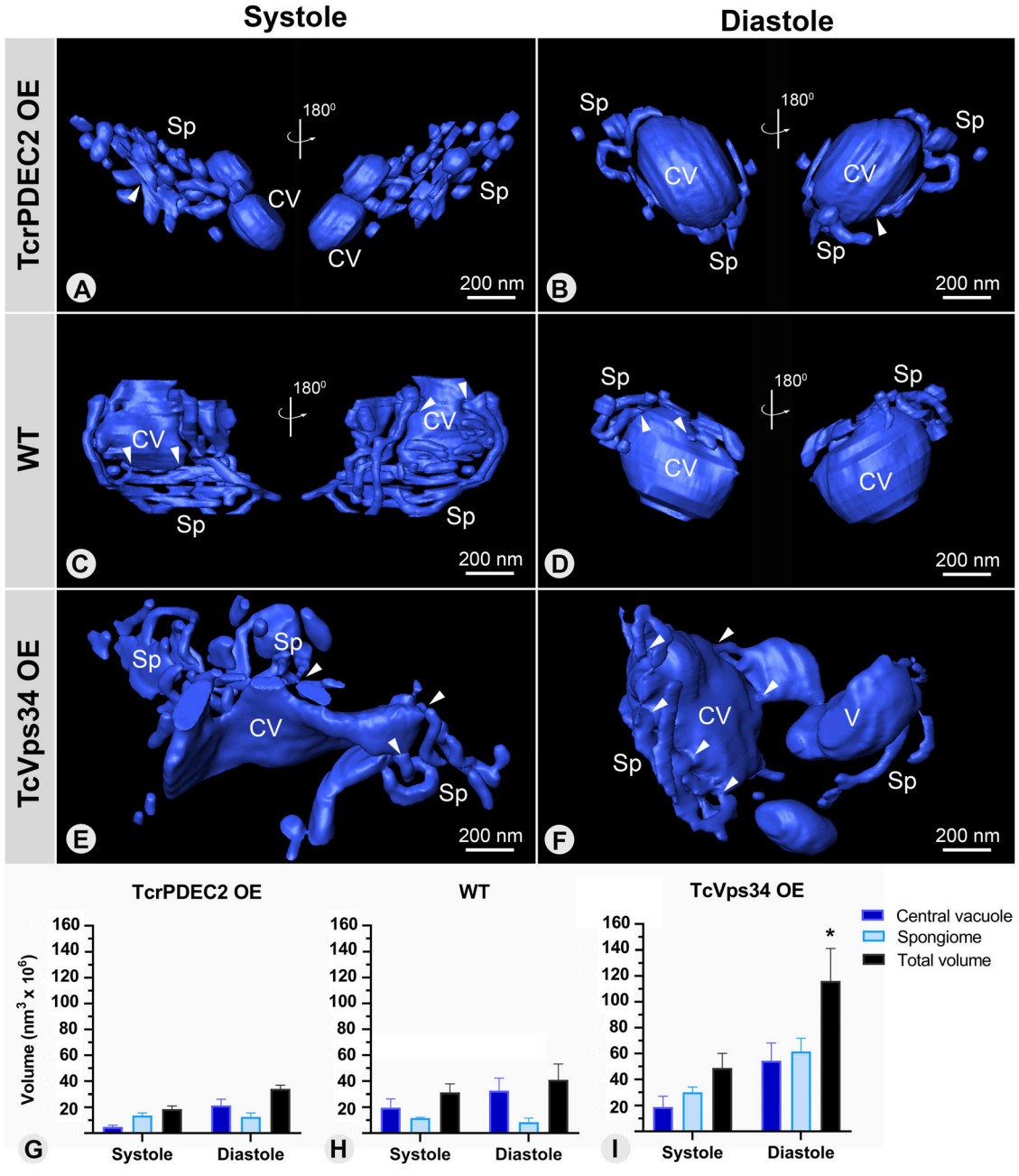

**Figure 5. Three-dimensional remodeling of CVC during systole and diastole stages.**
**(A, B, C, D, E, F)** 3D models of the CVCs obtained from HPF/FS single cells of TcrPDEC2 OE, WT, and TcVps34 OE cells in systole and diastole, respectively. The arrowhead points to connections between spongiome tubules and vesicles with the central vacuole. These models are representative of tomography datasets obtained from multiple cells in each condition. **(G, H, I)** Quantification of the volumes of the spongiome, central vacuole, and CVC's total volumes at the systole and diastole stages for TcrPDEC2 OE, WT, and TcVps34 OE, respectively. Data values are expressed as mean ± SEM, n = 5 per each experimental group. A one-way ANOVA test was applied. In (I), *$P$ ≤ 0.02 refers to the significant difference in the total volume of the CVC in WT in the diastole phase. CV, central vacuole; Sp, spongiome.

reduction in TcrPDEC2 OE cells from 80% during systole to 76% in diastole (Tables 2 and 3), In contrast, TcVps34 OE parasites showed nearly a 50% reduction in the proportion of small fragments, from 36% in systole to 18% in diastole. This significant decrease suggests a high frequency of fusion events within the CVC during pulsation in these cells (Tables 2 and 3). These findings show that TcrPDEC2 OE cells, which have a low-efficiency response to osmotic stress,

possess a fragmented spongiome with weak connections to the central vacuole. In contrast, TcVps34 OE mutants, which are superefficient responders, exhibited a well-developed and highly interconnected spongiome. This supports the critical role of the CV-spongiome structure and its remodeling during the CVC pulsation cycle in effectively responding to osmotic stress in *T. cruzi*. The CVC morphology in mutant cells also illustrates how alterations in the

**Table 2. Number of fragments forming the CVC at the systole stage categorized by size (surface area).**

| Fragment surface area (nm$^2$ x 10$^5$) | TcrPDEC2 OE | WT | TcVps34 OE |
|---|---|---|---|
| ≤0.5 | 87 | 21 | 13 |
| 0.5–1 | 21 | 2 | 7 |
| 1–10 | 12 | 1 | 15 |
| 10–20 | 0 | 4 | 0 |
| >20 | 0 | 0 | 1 |
| Total | 120 | 28 | 36 |

Quantification of fragments are expressed as absolute numbers, n = 5 per each cell line.

**Table 3. Number of fragments forming the CVC at the diastole stage categorized by size (surface area).**

| Fragment surface area (nm$^2$ x 10$^5$) | TcrPDEC2 OE | WT | TcVps34 OE |
|---|---|---|---|
| ≤0.5 | 64 | 20 | 6 |
| 0.5–1 | 10 | 3 | 5 |
| 1–10 | 9 | 6 | 20 |
| 10–20 | 1 | 1 | 0 |
| >20 | 0 | 0 | 1 |
| Total | 84 | 30 | 32 |

Quantification of fragments are expressed as absolute numbers, n = 5 per each cell line.

components of the osmoregulation signaling pathway can impact not only functionality and stress response but also the basic ultrastructural organization of the osmoregulatory system.

The volume of the CVC in both stages of the pulsation cycle was determined using three-dimensional models, providing clues on the water transfer dynamics along the pulsation cycle across the three genotypes. The CVC volume of the TcrPDEC2 OE cells was generally smaller when compared with WT parasites, whereas TcVps34 OE mutants exhibited a significantly larger CVC volume (Fig 5G).

During the systole stage, the average total volume of the CVC (the sum of the mean volumes of the central vacuole and spongiome) was 40% lower in TcrPDEC2 OE parasites compared with WT cells, whereas TcVps34 OE mutants exhibited average volume 58% higher (Fig 5G and Table S1). At the diastole stage, the average total volume for TcrPDEC2 OE cells was 17% less than that of WT parasites, and TcVps34 OE mutants had a volume 283% greater than that of WT parasites, a significant statistical difference that partially explains their physiological phenotypes (Fig 5G and Table S2).

To better understand the contribution of each component to the dynamics of water flow within the CVC, the volumes of the spongiome and the central vacuole were separately measured during the systole and diastole phases in the three genotypes (Fig 5G–I; Tables S1 and S2). The results showed that the spongiome volume of the TcrPDEC2 OE mutants did not change significantly between systole and diastole, increasing mainly the central vacuole volume by four times (Fig 5G). During systole, WT cells exhibited a central vacuole with an average volume slightly larger than the spongiome. When transitioning to diastole, the central vacuole expanded by ~65%, whereas the spongiome volume

decreased by about 25% (Fig 5H). TcVps34 OE mutants, in contrast, exhibited volume increases in both the spongiome and the central vacuole across the pulsation cycle stages, with the central vacuole expanding approximately three times its size in diastole (Fig 5I). The spongiome volume consistently exceeded that of the central vacuole in both phases (Fig 5I). This evidence points to the inability of TcrPDEC2 OE mutants to effectively capture water from the cytoplasm into the spongiome and transfer it to the central vacuole, whereas TcVps34 OE cells exhibited an expanded spongiome capable of capturing and transferring substantial water volumes, thereby enhancing its efficiency.

### The adhesion plaque integrates a dynamic membrane domain supported by a filamentous network

As described above, the CVC is consistently situated adjacent to the flagellar pocket (FP), forming a constant CVC-FP shared domain between their membranes. The characteristic electron density previously observed in ultrathin sections of the adhesion plaque (Linder & Staehelin, 1979; Girard-Dias et al, 2012), has been corroborated through electron cryotomography of unstained whole epimastigote cells (Fig 6A–D). In addition, 3D visualization of freeze-substituted samples through electron tomography revealed filaments that connect the membranes of the FP and the CVC in all genotypes examined (Fig 7). Whereas the intra-plate filaments were visible in the XY and YZ plans without a clear orientation pattern (Fig 7B, E, and H), they seem to constitute an organized filamentous network (Fig 7C, F, and I, Video 2). Statistical analyses showed significant differences in filament thickness and length across the groups. WT cells had 6 nm–thick filaments, whereas both mutant groups had

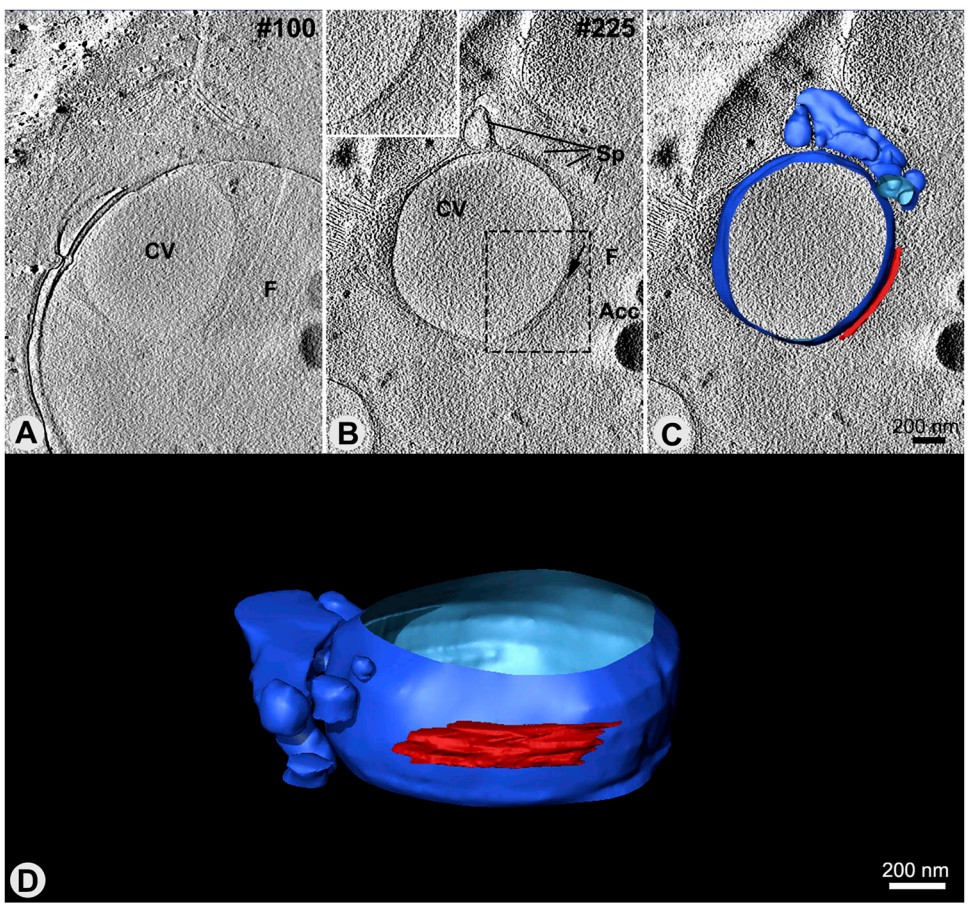

**Figure 6. The CVC and adhesion plaque observed by cryo-ET.**
**(A, B)** Virtual sections of the cryotomogram showing the proximity of the CVC to the flagellar pocket and their connection by the adhesion plaque (arrow). Here, the adhesion plaque is observed as an electron-dense domain even in the absence of stains. **(B, C)** Top view of the 3D model in the same section showed in (B). **(D)** 3D model showing partial volume of the central vacuole surrounded by the spongiome and the adhesion plaque (red) in the center of the central vacuole.

filaments ~8 nm thick (Fig 7J). Filament length in TcrPDEC2 OE cells was the longest, averaging 38 nm, in contrast to 27 nm in TcVps34 OE mutants and 20 nm in WT cells (Fig 7K). Such differences could suggest a larger distance between the central vacuole and the flagellar pocket in the TcrPDEC2 OE parasites or could be because of the orientation of most visible filaments, being obliquely positioned in this cell line.

Segmentation and 3D modeling of the interaction region between the CVC and the flagellar pocket (FP) revealed that in all genotypes, the membranes of the flagellar pocket and central vacuole were aligned side by side, displaying complementary shapes in the adhesion plaque region (Fig 8A–I and Video 3). The analysis of the 3D models revealed variability in their overall dimensions across all groups, with lengths varying from 75–450 nm and thicknesses typically between 10 and 30 nm. Notably, cells overexpressing TcrPDEC2 occasionally exhibited plaques with a thickness of 40 nm.

The dynamics of the adhesion plaque were also quantitatively assessed across the pulsation cycle. During the systole stage, all three genotypes displayed significant variation in the area of the adhesion plaque. Notably, TcrPDEC2 OE consistently showed the smallest surface areas ($\leq 6.6 \times 10^4$ nm$^2$), and TcVps34 OE mutants the largest ($\leq 13 \times 10^4$ nm$^2$), compared with WT cells ($\leq 9.2 \times 10^4$ nm$^2$) (Fig S2A). At diastole, the observed values presented a distinct scenario. Whereas a few WT cells exhibited

adhesion plaques with larger areas, the majority remained within the range observed during the systole stage (Fig S2A and B). Conversely, both mutants appeared to increase their adhesion plaque area during diastole (Fig S2B). Adhesion plaques of TcrPDEC2 OE cells showed areas $\leq 15 \times 10^4$ nm$^2$, comparable with or even exceeding those of WT cells, whereas most TcVps34 OE mutants had adhesion plaques of $\leq 18 \times 10^4$ nm$^2$, with a few cells reaching $\approx 40 \times 10^4$ nm$^2$ (Fig S2B).

To examine the relationship between changes in CVC structure and the dynamics of the adhesion plaque during the pulsation cycle, measurements of the plaque surface area were taken, alongside the volume data for the CVCs from Fig 5, revealing that changes in the size of the plaques are in sync with the fluctuations in CVC volume during the pulsation cycle. Similarly, WT cells did not show any change in the mean area of the adhesion plaque between systole and diastole stages, whereas both mutants doubled their adhesion plaque area following increasing volume of the central vacuole (Fig 8J). At the diastole stage, the TcVps34 OE cells showed a significant difference in the adhesion plaque area following changes in the CVC volume (Figs 5G and 8J). Thus, the adhesion plaque in *T. cruzi* is a dynamic structure that maintains the organization of the membrane domains around the CVC and the FP and can adapt to physiological and morphological changes, showing a synchronous organization with the CVC.

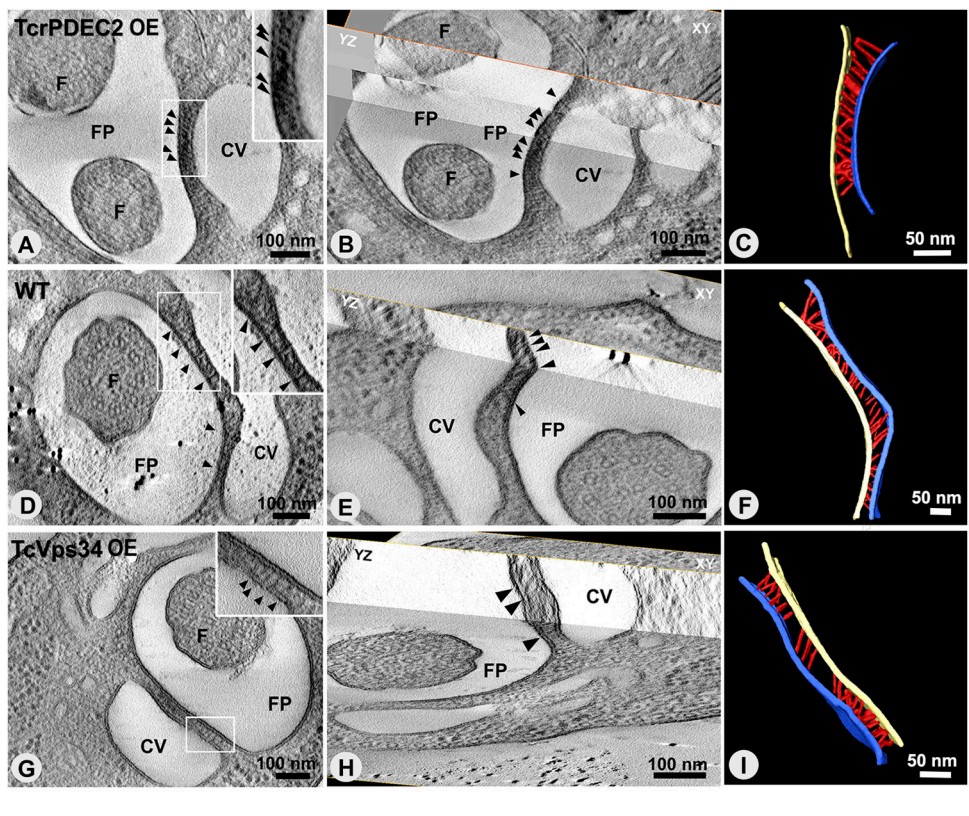

**Figure 7. The adhesion plaque contains filaments connecting the CVC and flagellar pocket membranes.**
**(A, D, G)** Virtual sections of tomograms showing the filaments inside the adhesion plaque domain. They are pointed by arrow heads. **(B, E, H)** Virtual sections of tomograms showing filaments also observed in the YZ plane with different orientations. **(C, F, I)** 3D models of the filamentous network from each genotype group. **(J, K)** Quantification of the thickness and length of the filaments observed in each genotype. Values are expressed as mean ± SEM, n = 40 filaments per each cell line. One-way ANOVA test applied. **P ≤ 0.005, ***P = 0.0002, ****P < 0.0001. CV, contractile vacuole; FP, flagellar pocket; F, flagellum.

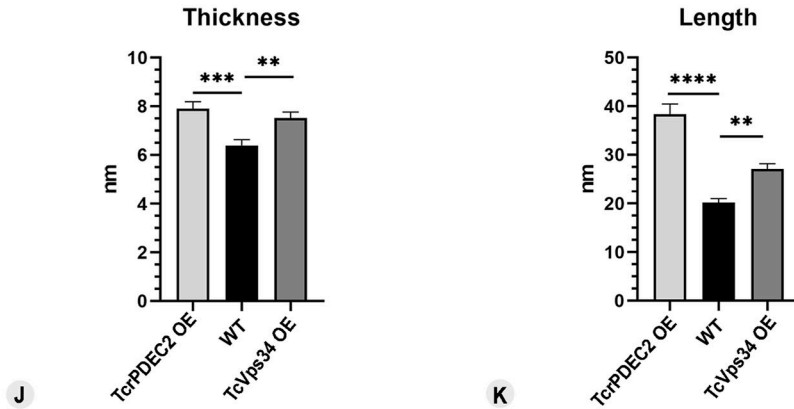

## Intramembranous particles (IMPs) and filamentous structures in the membranes of the CVC

Freeze-fracture assays revealed the inner face of the lipid bilayers located in the CVC-FP region. The standard structure of the CVC with the central vacuole adjacent to the flagellar pocket, surrounded by the spongiome, was easily observed. The replicas confirmed the structural phenotype of the CVC and the adhesion plaque (Fig 9A), as previously determined by ultrathin section analysis and 3D reconstruction. IMPs, randomly scattered across the central vacuole, spongiome, and acidocalcisome membranes, were observed (Fig 9B–E). These particles varied in size, measuring ~11 nm and between 7–9 nm in both the CVC and acidocalcisomes (Fig 9B–E).

The membranes of the central vacuole and the flagellar pocket were observed adjacent to each other, suggesting that at this contact site, both membranes form a functional domain that may regulate fluid discharge by the CVC (Fig 10A). Particle aggregates organized in rows (Fig 10B) and filamentous material (Fig 10C–E) were identified in the interaction region of these membranes, presumably representing the adhesion plaque area, as described in *L. collosoma*. A parallel alignment of the flagellar pocket and central vacuole membranes was also observed (Fig 10E). The connection of the central vacuole to the flagellar pocket through the adhesion plaque was evident in both systole (Fig 10D and E) and diastole (Fig 10A–C, F, and G) stages, especially near the base of the flagellar pocket (Fig 10D–G).

Collectively, our results indicate that the spongiome serves as the primary location for the initial absorption of water. A clear

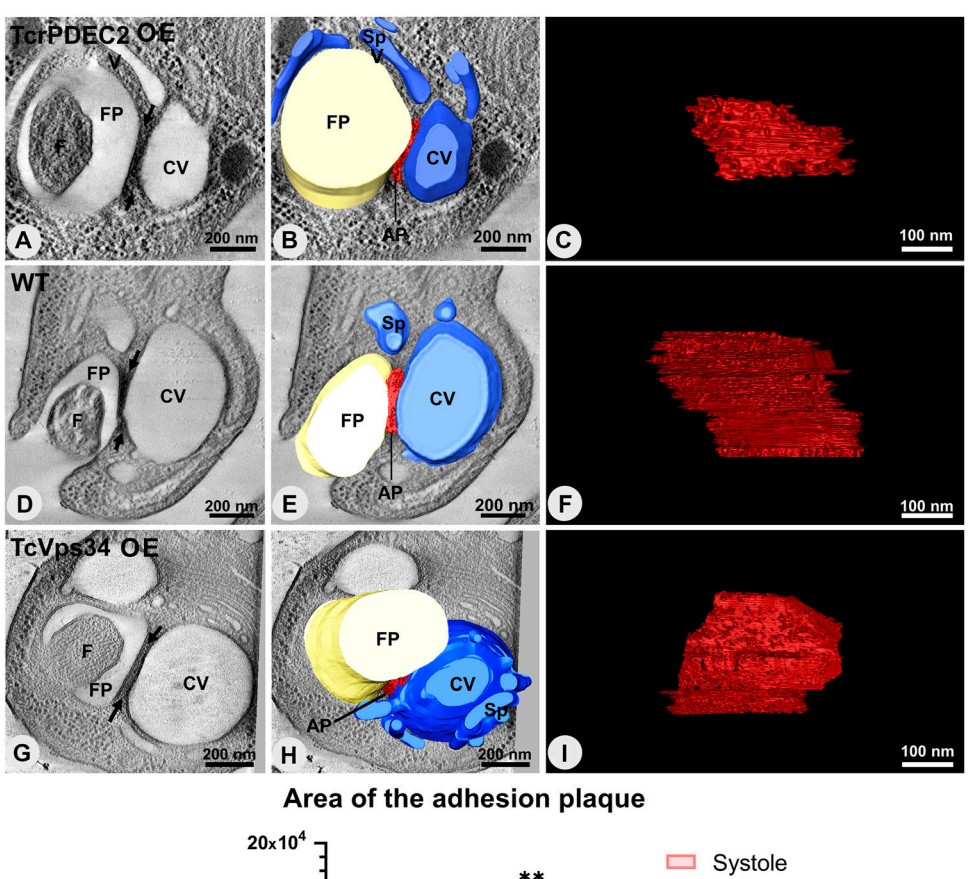

**Figure 8. The adhesion plaque structure has different sizes in TcrPDEC2 OE and TcVps34 OE mutants.**
**(A, D, G)** Virtual sections of tomograms showing the electrodense region of adhesion plaque between arrows. **(B, E, H)** Highlights of the top view of respective three-dimensional models from tomograms. **(C, F, I)** Showing a front view of the adhesion plaque 3D models. **(J)** Quantification of the adhesion plaque area in both phases of the CVC cycle pulsation. Values are expressed as mean ± SEM in nm$^2$, n = 5 per each experimental group. One-way ANOVA test applied. **$P$ = 0.004. CV, contractile vacuole; Sp, spongiome; FP, flagellar pocket; F, flagellum; AP, adhesion plaque.

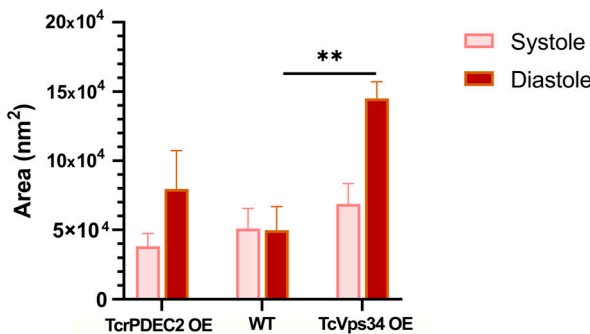

structural phenotype that explains the physiological response to osmotic stress was identified. Less efficient cells present fragmented spongiomes, whereas cells displaying highly efficient responses exhibit large and extensively connected CVCs. In addition, a novel membrane domain mediating a permanent contact between the flagellar pocket and the CVC was identified by electron tomography and FIB-SEM. Reconstructions that covered the complete CVC volume showed the presence of adhesion plaque in 100% of the cells, including the mutants. This newly identified domain undergoes morphological changes in tandem with the CVC during the pulsation cycle, increasing in size from systole to diastole. Superefficient mutants displayed larger adhesion plaques, suggesting their active role in osmoregulation. In addition, filaments connecting the central vacuole to the flagellar pocket membrane were observed in the adhesion plaque domain, presumably providing mechanical support and stability to this domain in trypanosomatids.

## Discussion

Osmoregulation plays a central role in cell biology, being crucial for a wide range of physiological processes, including kidney function, tissue organization, inflammation response, and blood cell integrity (Kültz, 2001; Bourque, 2008; Mavrogonatou & Kletsas, 2009; Brocker et al, 2012). In the context of trypanosomes, osmoregulation assumes a particularly significant role as it shields the parasite from osmotic stress encountered along its biological cycle and potentially enhances its capacity to resist antiparasitic drugs (Rohloff et al, 2004). Many antiparasitic agents exert their effects by compromising the integrity of cellular membranes, triggering morphological changes that often result in rounded shape parasites (Raja et al, 2017; Sulik et al, 2023). Thus, understanding osmoregulatory mechanisms in trypanosomes is key for developing strategies to counter their resistance and adaptability.

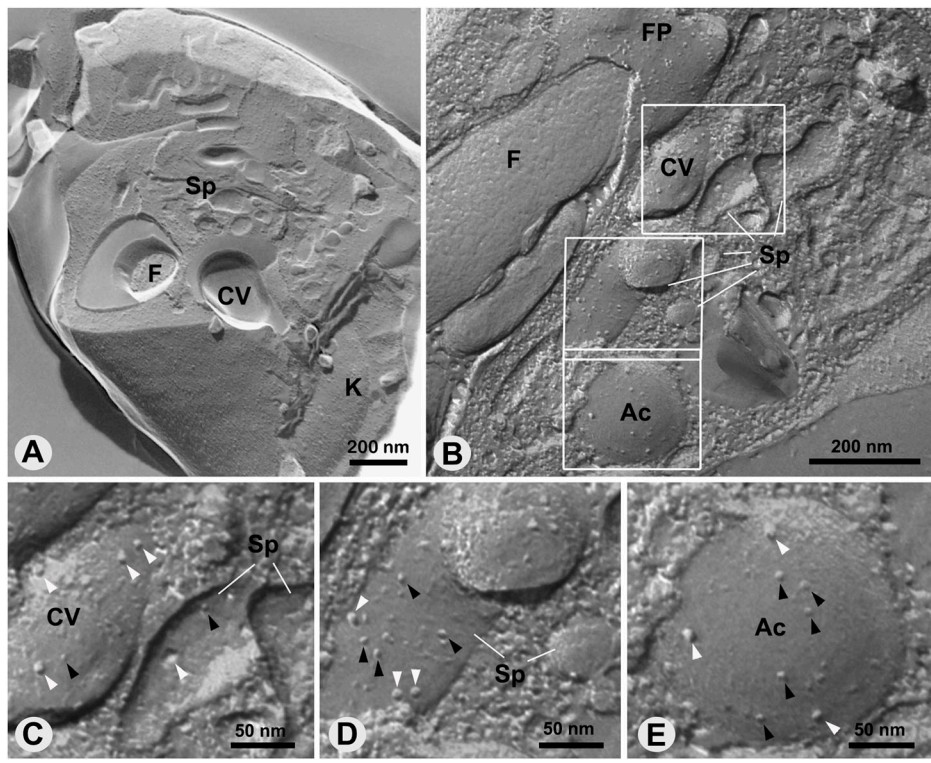

**Figure 9. Distribution of Intramembranous particles across the CVC-FP region revealed by freeze-fracturing.**
**(A, B)**, images of replicas of anterior region of the parasite showing the central vacuole, the spongiome, the flagellar pocket, the flagellum, the kinetoplast, and an acidocalcisome. **(B, C, D, E)** corresponding selected areas in (B) where Intramembranous particles can be observed randomly distributed. Large particles measuring 11 nm (white arrowheads) and between 9 and 7 nm (black arrowheads) were observed. FP, flagellar pocket; F, flagellum; CV, central vacuole; Sp, spongiome; K, kinetoplast; Ac, acidocalcisome.

## Spongiome fusion and the dynamic remodeling of CVC organization during osmoregulation

The use of ultrarapid cryofixation techniques was important for the observation of structures in their close-to-native state, revealing the dynamic changes on the structure of the contractile vacuole. Three-dimensional analysis revealed the general morphology of the CVC across the three developmental stages of *T. cruzi* and pinpointed the different phases in the dynamic process of remodeling, providing the first quantitative description of the dynamic changes in the CVC along its pulsation cycle.

In other models, the spongiome typically functions in the uptake and formation of water-collecting ducts that transport water to the central vacuole (Patterson, 1980; Allen & Fok, 1988; Allen, 2000; Allen & Naitoh, 2002). In *T. cruzi*, the spongiome exhibited tubules with a wide range of diameters and projections to long distances, suggesting that a substantial initial water uptake occurs via the spongiome. The water is then transferred via direct fusion of spongiome tubules and vesicles with the central vacuole, without the formation of a well-defined duct or channel. Finally, successive fusion events contribute to an increase in the volume of the central vacuole because of distention (i.e., stretching or expansion when changing from lamellar to spherical shape) and incorporation of membranes.

Events of volume and surface area expansion were previously discussed by Groulx et al (2006), who showed in various cell models that the increase in cell volume because of hypoosmotic stress is primarily attributed to conformational changes, distension of irregular membranes, incorporation of membranes through vesicle fusion, and to a lesser extent, tension in the phospholipid bilayer (Groulx et al, 2006). The larger average volumes observed in spongiomes during diastole (Fig 5) could be explained by one or a combination of the following scenarios: (i) they are in intermediate stages of diastole, and were not completely fused with the CVC; (ii) the fusion of vesicles from the spongiome to the central vacuole occurs simultaneously with the volume increase (i.e., water uptake) and/or surface area expansion (e.g., fusion) of its components, (iii) the initial expansion of the central vacuole may stem from the unfolding of its own membrane. It has been suggested that for moderate changes in cell volume, the increase in membrane surface area occurs primarily because of the unfolding of pre-existing membranes, whereas fusion and membrane insertion events play a smaller role (Groulx et al, 2006). This would explain the minor differences observed in the quantification and classification of WT CVC fragments between systole and diastole (Tables 2 and 3), given that the analyses were conducted under isosmotic conditions because *T. cruzi* does not exhibit a synchronized pulsation cycle. Likewise, it is possible that the volume increase in the CVC in *T. cruzi* primarily results from the conformational changes of the central vacuole, transitioning from lamellar to spherical morphology by unfolding and incorporation of membranes via fusion with spongiome tubules and vesicles.

The differences in the CVC morphology in TcrPDEC2 OE and TcVps34 OE mutants may also explain their physiological responses and their respective low and high efficiency in osmoregulation. The overexpression of TcVps34 resulted in pronounced alterations in the morphology of both central vacuole and spongiome, exhibiting a considerably larger central vacuole and a large and highly connected spongiome. These characteristics enhance their ability to collect water, accounting for the improved response under hypoosmotic stress (Schoijet et al, 2008). The interconnection

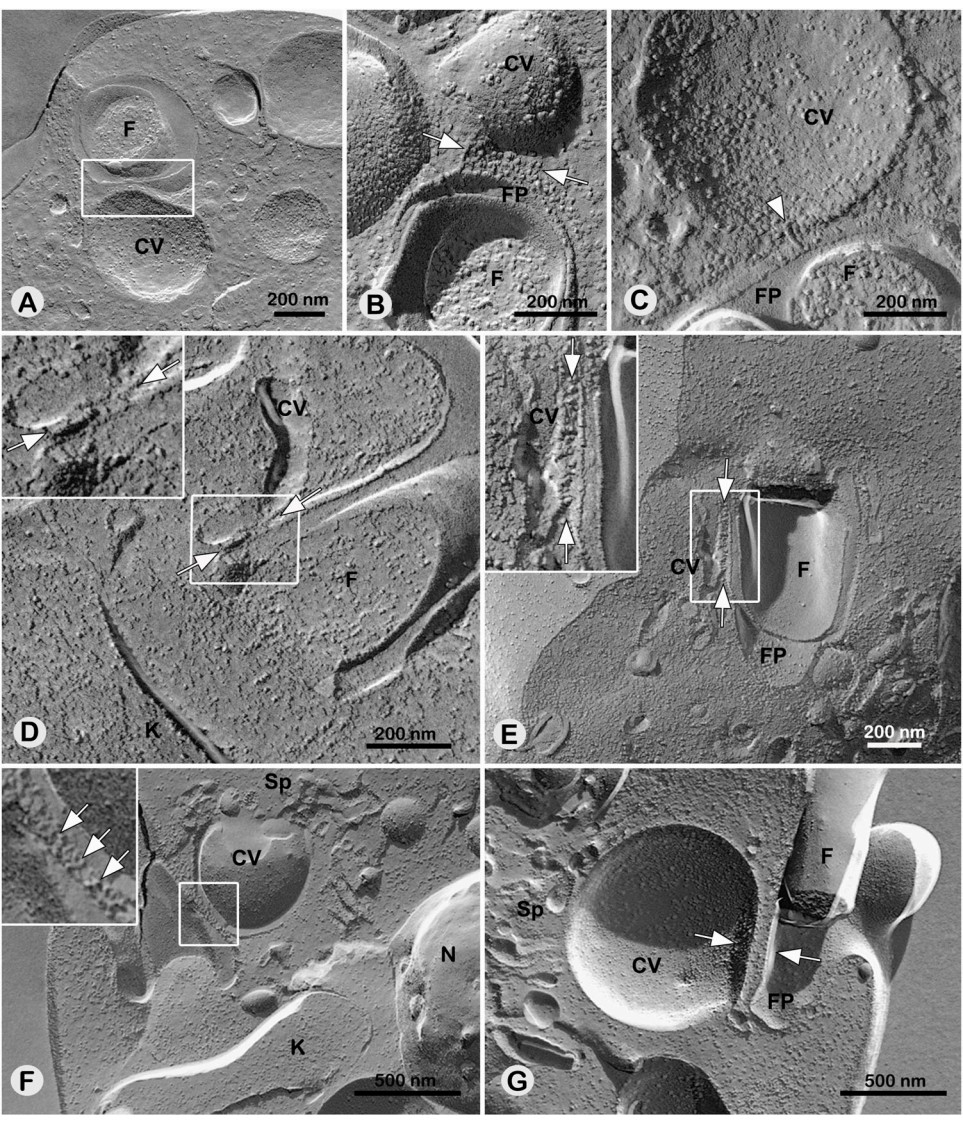

**Figure 10. The membrane domain of the attachment zone between the central vacuole and the flagellar pocket is rich in intramembranous particles.**
**(A, B, C, D, E)** WT; (F, G) TcVps34 OE mutant. **(A)** Highlighted in square the membranes of CV and flagellar pocket set side by side. **(B)** Row organization of particles in plaque adhesion pointed by arrows. **(C)** Arrowhead indicates filamentous IMPs linking the flagellar pocket and the central vacuole membranes. **(D, E)** Central vacuole in lamellar shape connecting to the flagellar pocket by particles pointed by arrows in the adhesion plaque (insets). **(F)** Arrows show a set of IMPs in the adhesion plaque (inset). **(G)** Arrows point to the parallelism of the flagellar pocket and the CVC membranes. FP, flagellar pocket; F, flagellum; CV, central vacuole; Sp, spongiome; K, kinetoplast; N, nucleus; arrows indicate the adhesion plaque, (*) points to parallelism of flagellar pocket and CVC membrane.

between the spongiome and the central vacuole, therefore, plays a critical role in the capacity of the CVC to accumulate water.

Although the overall CVC volume in TcrPDEC2 OE mutants was found to be similar to that of WT cells, this mutant exhibited a notably fragmented CVC, characterized by limited interconnections and reduced fusion with the central vacuole, which may explain their lower RVD response during osmotic stress. This structural phenotype is similar to what has been described in *Amoeba proteus* and *Chlamydomonas reinhardtii*, in which the CVC breaks into multiple vesicles and tubules, following water elimination to the external environment to coalesce once more during the diastole phase and form the central vacuole (Luykx et al, 1997; Allen & Naitoh, 2002). It is known that proteins that express a FYVE domain, such as PDEC2, play critical role in vesicle trafficking and fusion in different organisms (Gillooly et al, 2001). Concurrently, reduced levels of cAMP have been implicated in the diminished activity of aquaporin and restricted mobility within the CVC structure, potentially leading to less fusion events and

less efficiency in RVD (Kuwahara et al, 1995; García et al, 2001; Rohloff et al, 2004). Interestingly, localization of receptor-type adenylate cyclase and cAMP response protein 3 revealed the existence of two distinct signaling microdomains in *T. cruzi* (Chiurillo et al, 2023), one of them around the CVC, suggesting that responses to local fluctuations in cAMP might play a significant role in the parasite physiology.

## Freeze fracture shows that *T. cruzi* does not contain a decorated spongiome

Most protozoa possessing a CVC display a decorated spongiome, characterized by organized IMPs ranging from 11–15 nm in size. These particles are usually observed in freeze-fracture replicas and as an electron-dense outer layer in ultrathin sections (Allen & Naitoh, 2002). In *Paramecium*, spongiomes are distinguished into two types—smooth and decorated—and have been described based on the arrangement of IMPs, suggesting the existence of

different functional compartments. The IMPs on the decorated tubules of the CVC of Paramecium has been proposed as the proton pump VH+-ATPase (Fok et al, 1995; Allen, 2000).

Freeze-fracture replicas revealed IMPs in *T. cruzi* with a maximum size of 11 nm, randomly distributed across the spongiome, the central vacuole, and acidocalcisome surfaces. The particles found in the spongiome might correspond to clusters of aquaporin 1 (TcAQP1), previously localized in the CVC (Montalvetti et al, 2004), which may explain the presumed increase in water flow to this compartment. Other candidate proteins which have been also localized to the CVC membrane are the vacuolar proton pyro-phosphatase (TcPPase or TcVP1), calmodulin (Montalvetti et al, 2004), alkaline phosphatase (Rohloff et al, 2004), and a poly-amine transporter (TcPOT1) (Hasne et al, 2010). A proteomic study has revealed the presence of SNARE 2.1 and SNARE 2.2, VAMP 7, AP180 and GTPases Rab11 and Rab32 proteins in CVC membrane (Ulrich et al, 2011; Niyogi et al, 2015). These proteins are considered essential for osmoregulation in *T. cruzi* and may play significant roles in the fusion events involving vacuole membranes and acidocalcisomes, suggesting the participation of the CVC in transporting GPI-anchored proteins to the plasma membrane (Niyogi et al, 2014; Niyogi & Docampo, 2015). Because evidence for the two types of spongiome—smooth and decorated—was not found in *T. cruzi*, it is likely possible that the spongiome in this parasite may primarily constitute a single functional domain.

### The adhesion plaque as a dynamic docking microdomain in *T. cruzi*

The first description of a specialized docking region connecting the central vacuole to the flagellar pocket in trypanosomatids was made in *L. collosoma* (Linder & Staehelin, 1979). An electrodense region, ~25 nm thick with 300 nm of diameter, filled with fibrillar material, was observed throughout the entire pulsation cycle of the CVC. Since this initial observation, there has been no further in-vestigation or progress in understanding this domain and its functionality in other trypanosomatids. In a previous work, we confirmed the presence of the adhesion plaque in *T. cruzi* (Girard-Dias et al, 2012). Here, using freeze-fracturing and serial electron tomography, the structure of the adhesion plaque in *T. cruzi* was dissected, revealing how it interacts with the CVC during its pul-sation cycle. As observed in *L. collosoma*, the adhesion plaque in *T. cruzi* features an electrodense domain ~30 nm thick, containing a filamentous network. The thickness of the filaments in the adhesion plaque aligns closely with other filamentous structures previously identified in the parasite, including the connection of subpellicular microtubes to the plasma membrane (~6 nm), filaments of the paraflagellar rod—PFR (7 nm), and filaments linking the axoneme to PFR domain (7 nm) (Souto-Padrón et al, 1984; Farina et al, 1986; De Souza, 2009; Rocha et al, 2010), suggesting that they may share similar connection characteristics. Although the filaments of TcVps34 OE and TcrPDEC2 OE mutants seemed to be longer and slightly larger than those in WT cells, no significant differences were found in their general morphology among these groups. This fil-amentous network likely plays a role not only in maintaining the proximity of the central vacuole to the flagellar pocket but also in providing mechanical stability to their membranes. Indeed,

artifacts created by shear forces during sample preparation for cryo-EM and HPF showed that, whereas the regions adjacent to CVC and FP are disrupted, their shared domain remains intact (Fig S3A–E). In addition, both *T. cruzi* and *L. collosoma* exhibit a par-allel alignment of the flagellar pocket and CVC membranes within the adhesion plaque domain (Linder & Staehelin, 1979), supporting the idea that the adhesion plaque contributes to the stabilization of their membranes. This arrangement might, therefore, imply the potential existence of a specific, shared microdomain on the membranes of the central vacuole and the flagellar pocket, opening questions on the potential functional role or influence of the adhesion plaque on vesicle trafficking via the flagellar pocket. This has also been reported in other flagellates and amoeba, suggesting the conservation of such domains across different evolutive groups (Patterson, 1980). A transient electrodense region with filaments connecting the contractile vacuole to the site of discharge was described in *Mesostigma viride*, the only flagellated organism in the green algae group Streptophyta (Buchmann & Becker, 2009). Nevertheless, unlike *M. viride*, *T. cruzi* and *L. collosoma* display a permanent adhesion plaque along the entire pulsation cycle. Al-though the biogenesis of the CVC in protists is still not completely understood, the existence of a permanent docking domain in the contractile vacuole of trypanosomatids may also impact its for-mation. It is possible that it serves as a nucleation site for de novo CVC biogenesis, as hypothesized for ciliates (Plattner, 2013, 2015).

Characterization of the adhesion plaque in TcrPDEC2 OE and TcVps34 OE mutants also showed how dynamically organized this domain is, duplicating its entire area from systole to diastole. Such changes are likely associated with the dynamic changes observed in the CVC of the mutants, whereas WT cells showed just moderate structural changes during the pulsation cycle. It could also po-tentially represent the recruitment of prefabricated elements of the adhesion plaque to follow the CVC changes under osmotic stress.

A similar transient electron-dense domain was observed in swollen contractile vacuoles of *C. reinhardtii* (Weiss et al, 1977), which does not have a permanent central vacuole (Luykx et al, 1997). Freeze-fracture replicas of *C. reinhardtii* also showed a circular plaque on the surface of the contractile vacuole, leading the au-thors to theorize this structure as a gate for water efflux (Weiss et al, 1977). Later, analysis of ultrathin sections by EM provided evidence for the formation of a fusion pore. A single image containing a pore of ~15 nm in the center of the electron-dense contact zone between an expanded contractile vacuole and the plasma membrane was shown (Luykx et al, 1997).

The precise mechanism of fluid discharge from the CVC during osmotic stress is still unknown. Here, thin sections revealed a 25 nm gap in the adhesion plaque in cells at their final stage of diastole, likely representing a site for pore formation. Round perforations of 20–40 nm were also observed in the adhesion plaque and adjacent membranes of *L. collosoma* in freeze-fracture replicas. Neverthe-less, membrane fusion events were not seen, and the authors proposed a model of membrane rupture as a mechanism for fluid discharge in trypanosomatids (Linder & Staehelin, 1979). Consid-ering the energy cost of rupturing and remaking of the phospholipid bilayer and the lack of control of such mechanisms, it is unlikely that it takes place multiple times under physiological conditions (Jimenez et al, 2014; Rodenfels et al, 2020; Ammendolia et al, 2021).

Considering that membrane fusion events have a half-life of ~15 milliseconds (Reinhard et al, 2003; Eyring & Tsien, 2018; Sharma & Lindau, 2018), the lack of data on the contractile vacuole discharge mechanism is understandable and elucidating it will require high resolution, fast, in vivo analysis.

In summary, by using cells with different physiological response to osmotic stress, we were able to correlate the morphological and physiological changes in different phases of the CVC pulsation cycle with unprecedented resolution. The importance of the spongiome structure throughout the remodeling of the CVC was evident. Imaging the adhesion plaque domain at high resolution also provided insights into its functional role in the fluid discharge during osmoregulation in this parasite.

# Materials and Methods

### Cell culture

*T. cruzi* epimastigote forms (CL Brenner strain) were cultivated in liver infusion tryptose medium as previously described (Camargo, 1964) supplemented with 10% (vol/vol) FBS at 28°C. Mutants overexpressing PI3K (TcVps34 OE) (Schoijet et al, 2008) and PDEC2 (TcrPDEC2 OE) (Schoijet et al, 2011) were cultivated as described above, under pression selection of 500 $\mu$g/ml of G418 (Invitrogen). They were collected after 3 d of culture.

### Cell extracts and western blot analysis

For *T. cruzi* extracts, $1 \times 10^8$ epimastigotes from TcrPDEC2 OE and TcVps34 OE tagged with HA were harvested by centrifugation at 1,500 g for 10 min and washed twice with PBS. Cell pellets were then resuspended in lysis buffer (50 mM HEPES buffer, pH 7.3, 0.01 mg/ml leupeptin, 25 U/ml aprotinin, 0.5 mM phenylmethylsulfonyl fluoride, and 14 mM 2-mercaptoethanol) and lysed by six cycles of freezing in liquid N2 and thawing at 4°C. The total extracts were further centrifuged for 1 h at 100,000$g$ to obtain P100 and S100 fractions.

For Western blot analysis, proteins were resolved in 8% (wt/vol) SDS-polyacrylamide gel electrophoresis as described by (Laemmli, 1970) and electrotransferred to Hybond-C membranes (Amersham Pharmacia Biotech). The membranes were blocked with 5% (wt/vol) non-fat milk suspension in TBS-Tween for 2 h. For TcrPDEC2, an incubation overnight was performed with 1:1,000 dilution of the rabbit anti-TcrPDEC2 antiserum obtained as described in Schoijet et al (2011), and detection was carried out by incubating with a 1:5,000 dilution of a goat anti-rabbit IgG labelled with peroxidase (KPL). For TcVps34, the transferred membranes were incubated for 2 h with a 1:4,000 dilution of rat anti-HA high affinity monoclonal antibody (Roche Applied Science), and detection was carried out by incubating with a 1:5,000 dilution of a goat anti-rabbit or goat anti-rat conjugated to peroxidase (Sigma-Aldrich). The membranes were then developed with the ECL PlusTM Western Blotting Detection System (NEN Life Science Products). To control for sample loading, the blots were also probed with a monoclonal mouse anti–$\beta$-tubulin antibody (1:400 dilution; Chemicon International).

### Hypoosmotic treatment

For the induction of hypoosmotic stress, epimastigotes forms were washed and resuspended in a standard isosmotic solution containing 116 mM NaCl, 5.4 mM KCl, 0.8 mM MgSO4, 5.5 mM glucose, 50 mM of Hepes, pH 7.4, with a final osmolarity of 300 mOsm. The final osmolarity was adjusted using the Osmomette A (Precision Systems Inc.). The hypoosmotic stress (150 mOsm) was induced by diluting 1:1 of the cell suspension in the standard solution with Milli-Q water (Millipore) as established by Rohloff et al (2004).

### Freeze-fracturing

*T. cruzi* epimastigotes cells from WT and overexpressing-TcVps34 strains were fixed for 1 h in 2.5% glutaraldehyde in 0.1 M cacodylate buffer at room temperature and stored in cacodylate buffer. Glutaraldehyde-fixed trypanosomes were infiltrated in 15% glycerol for 4 h and further incubated in 30% glycerol overnight. The cells were pelleted by centrifugation and frozen in the liquid phase of partially solidified Freon 22 cooled by liquid nitrogen. Freeze fracture was carried out at −140°C in a BAF 060 (Bal-Tec) apparatus. The specimens were shadowed from a platinum-carbon (Pt/C) source at $2 \times 10^{-5}$ Torr within 2–5s of fracturing. Replicas were collected and cleaned in 1% sodium hypochlorite and Milli-Q water for 24 h each. They were observed using the transmission electron microscopy Jeol 1200 EX at 80 kV.

### HPF

For HPF, cells in liver infusion tryptose medium were centrifuged at 1,500$g$ for 5 min and the pellet was sandwiched between aluminum carries (0.5 mm) (Leica Microsystems). The pellet was placed between two types of carries (type A and B) so that the cells were protected in a 200 mm cavity on one carrier. The pellet was also inserted by capillarity in 3 mm pieces of cellulose capillaries and one end of the capillary was closed using tweezers. Four capillaries were mounted between the two carries in the same way described above. In this case, the cavities were filled with hexadecane to avoid air between the capillaries. The sandwiched samples were mounted in the HPF holder and frozen using a Bal-Tec HPM 010 HPF machine (Bal-Tec, Corp.).

### Freeze substitution and resin embedding

Samples were carefully removed from the liquid nitrogen and immersed in the substitution medium (2% osmium tetroxide, 0.1% glutaraldehyde, and 1% of water in acetone), pre-cooled to −90°C using a Leica AFS2 apparatus (Leica Microsystems). The samples were kept at −90°C for 72 h, then slowly increased 4°C per hour until −20°C. After 2 h, it was heated to 4°C. After substitution, samples were washed three times with acetone at room temperature and then stepwise embedded in Epon (PolyBed 812) and polymerized at 60°C for 72 h.

 **Life Science Alliance**

### Electron tomography (ET)

For ET, ribbons of serial sections of 200 and 300 nm from high-pressure frozen and freeze-substituted resin embedded samples were collected on Formvar-coated slot copper grids. Samples were post-stained with uranyl acetate and lead citrate and incubated with 10 nm colloidal gold on both sides for 10 min.

Tomograms were acquired on a Tecnai G20 scanning-transmission electron microscope (Thermo Fisher Scientific) operating at 200 kV accelerating voltage on the STEM mode using a HAADF detector and on TEM mode using a 4k x 4k CMOS camera (AMT). Tilt series from –65° to +65° with an angular increment of 2° were used to acquire all tomograms.

### FIB microscopy

For observation by FIB-SEM, embedded samples were trimmed to a trapezium shape, and the block surface was smoothed by sectioning using a conventional diamond knife. The block was then glued to an SEM stub using silver glue, with the smooth surface facing upwards, perpendicular to the microscope column, covered by 10 nm of gold. Samples were imaged using a Auriga 40 cross-beam microscope (Zeiss) equipped with a gallium-ion source for focused ion-beam milling. The cross-sectional cut was made at ion beam currents of 30 kV:50pA and images were acquired with an electron beam of 1.8 k, dwell time of 1.5 min; totalizing about 15–25 µm. A series of backscattered electron images were recorded in "slice-and-view" mode, at a milling step size of 30 nm, magnifications between 10 and 16 K, with a pixel size of 4.9, 5.6, and 8.9 nm for WT, TcVps34 OE, and TcrPDEC2 OE cells, respectively.

### Cryoelectron microscopy and cryoelectron tomography

WT cells were washed and resuspended in an isosmotic buffer (116 mM NaCl, 5.4 mM KCl, 0.8 mM $MgSO_4$, 5.5 mM glucose, and 50 mM HEPES; pH 7.4) (Rohloff et al, 2004). The osmolarity was adjusted to 300 mOsm using the Osmomette A (Precision Systems Inc). The sample (3 µl) was pipetted onto lacey carbon (200 mesh; Pelco) and 200-mesh R 2/2 copper Quantifoil grids previously prepared by glow discharge (EasyGlow; Pelco). The grids were frozen by plunge-freezing in liquid ethane using a Vitrobot Mark III system (Thermo Fisher Scientific). Conditions were set to 28°C and 100% humidity, following a blotting force of 4 and a blotting time of 5s. Cryo single images were acquired at a Tecnai G20 transmission electron microscope (Thermo Fisher Scientific) operating at 200 kV and equipped with a Thermo Fisher Eagle digital camera with a detector of 2048 × 2048 pixels. Tomographic data were collected using the Tomo package on a Titan Krios G3i (Thermo Fisher Scientific) equipped with Falcon 3EC. Samples were tilted every 3 degrees to ± 50° using a dose-symmetric scheme under a total dose of 100 e−/Å2, at 8,700× magnification with a defocus of −13. Single images and virtual section contrasts were corrected with the Enhance Local Contrast filter (CLAHE) from Fiji/Image J software (Schindelin et al, 2012).

### 3D reconstruction and visualization

Both tomographic tilt series and image series were processed using version 4.9.13 of IMOD (University of Colorado, USA) (Kremer et al, 1996). Projections were aligned by cross-correlation with final alignments carried out using 10 nm fiducial gold particles followed by weighted back-projection reconstruction. For segmentation and data display, the reconstructed volumes were processed by a Gaussian filter. Manual and semi-automatic segmentations and surface rendering were done with AMIRA (Visage Imaging).

### Morphometric and statistical analyses

The morphometric parameters of the central vacuole were determined from ultrathin sections (70 nm) of epimastigote cells, submitted or not to hypoosmotic treatment. The central vacuole profiles of 30 cells in each osmotic condition (iso- and hypoosmotic) were segmented and measurements were acquired using the Image J program (NIH, USA) (Schindelin et al, 2012). Statistical significance was determined by $t$ test.

The quantification of the volume and continuous area of the CVC was measured from 3D models of cells using the volume per slice and surface area tools from AMIRA (Visage Imaging). The complete volume of CVCs was acquired from five cells at systole and diastole in each group, totaling 30 cells analyzed. The volume results were expressed by mean ± SEM. The continued area of the 3D models was used to determine the fragment size and quantity in the five CVCs of each experimental group. The statistical significance was determined by a one-way ANOVA test.

The surface area was measured from 3D models of adhesion plaques from FIB image sequences and electron tomograms. Nine to 15 models were obtained per each experimental group. Because of the high variability found in this structure, the mean ± SEM and statistical test were applied only in the models from the cells selected to calculate the CVC measurements. The statistical significance was determined by a one-way ANOVA test.

The thickness and length of filaments found inside the adhesion plaque domain were measured using Ferret's diameter tool from the software Image J (NHI, USA) on virtual sections of peripheral and central regions of plaques from a total of 40 filaments per each genotype. The results were expressed by mean ± SEM. The statistical significance was determined by one-way ANOVA test. To all, tests were considered $P < 0.05$ at 95% of confidence interval (CI) to be significant.

### Online supplemental material

Three supplemental figures include additional data. Fig S1 contains the Western blotting confirming the overexpression of the proteins PDEC2 and Vps34. Fig S2 contains the graphs of distribution showing the variability of area of the adhesion plaque domain. Fig S3 contains images showcasing adhesion plaque's filaments structural stability even in the presence of cryofixation artifacts. Video 1 show in detail the 3D model of the CVC during hypoosmotic condition. Video 2 shows the 3D model of the filamentous network observed inside the adhesion plaque domain. Video 3 displays the interaction between CVC, adhesion plaque and flagellar pocket. Tables S1 and S2 provide the values for the average volume of central vacuole and spongiome. Table S3 describes the average of the adhesion plaque area.

## Supplementary Information

## Acknowledgements

We thank all the staff of CENABIO/UFRJ and LNNano/CNPEM (20230573) facilities for technical assistance. This work was supported by Conselho Nacional de Desenvolvimento Científico e Tecnológico (CNPq), Fundação Carlos Chagas Filho de Amparo à Pesquisa do Estado do Rio de Janeiro (FAPERJ), Financiadora de Estudos e Projetos (FINEP), Ministério da Saúde (Brazil) and Coordenação de Aperfeiçoamento de Pessoal de Nível Superior (CAPES) (Brazil). The authors declare no competing financial interests.

### Author Contributions

I Augusto: data curation, formal analysis, investigation, methodology, and writing—original draft, review, and editing.
W Girard-Dias: data curation, formal analysis, investigation, methodology, and writing—original draft.
A Schoijet: resources, data curation, formal analysis, investigation, and methodology.
GD Alonso: resources, formal analysis, funding acquisition, and methodology.
R Portugal: formal analysis and methodology.
W de Souza: resources, formal analysis, funding acquisition, project administration, and writing—review and editing.
V Jimenez: formal analysis and writing—review and editing.
K Miranda: conceptualization, resources, formal analysis, supervision, funding acquisition, methodology, project administration, and writing—original draft, review, and editing.

### Conflict of Interest Statement

The authors declare that they have no conflict of interest.

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
