## [Reviewer comments · Life Science Alliance]

Life Science Alliance

Quantitative assessment of the nanoanatomy of the contractile vacuole complex in *Trypanosoma cruzi*

Ingrid Augusto, Wendell Girard-Dias, Alejandra Schoijet, Guillermo Daniel Alonso, Rodrigo Portugal, Wanderley de Souza, Veronica Jimenez and Kildare Miranda

DOI: <https://doi.org/10.26508/lsa.202402826>

Corresponding author: Dr. Kildare Miranda (Federal University of Rio de Janeiro)

Review Timeline:

Submission Date:	2024-05-15
Editorial Decision:	2024-07-03
Revision Received:	2024-07-10
Editorial Decision:	2024-07-15
Revision Received:	2024-07-17
Accepted:	2024-07-18

Transaction Report:

July 3, 2024

Re: Life Science Alliance manuscript #LSA-2024-02826-T

Dr. Kildare Miranda
Federal University of Rio de Janeiro
Biophysics Institute
Av. Carlos Chagas Filho, 373, Bl C
Cidade Universitária
Rio de Janeiro, Rio de Janeiro 21941902
Brazil

Dear Dr. Miranda,

Thank you for submitting your manuscript entitled "Quantitative assessment of the nanoanatomy of the contractile vacuole complex in *Trypanosoma cruzi*" to Life Science Alliance. The manuscript was assessed by expert reviewers, whose comments are appended to this letter. We invite you to submit a revised manuscript addressing the Reviewer comments.

Thank you for this interesting contribution to Life Science Alliance. We are looking forward to receiving your revised manuscript.

Sincerely,

Eric Sawey, PhD
Executive Editor
Life Science Alliance
<http://www.lsa-journal.org>

B. MANUSCRIPT ORGANIZATION AND FORMATTING:

Reviewer #1 (Comments to the Authors (Required)):

The Authors report the morphological changes occurring in the contractile vacuole complex (CVC) of different life stages of *T. cruzi* as examined by different electron microscopy techniques (freeze fracturing, high pressure freezing, electron tomography, focused ion beam microscopy). They were also able to identify the changes occurring under hypoosmotic conditions, characterize the adhesion plaque that links the central vacuole to the flagellar pocket, and the changes occurring in the spongiome of wild type cells and cells overexpression a phosphodiesterase (TcPDEC2) and a kinase (TcVps34). This is a technically well executed study. I have a few questions/suggestions.

1. The Materials and Methods section indicates that hypoosmotic conditions were used only with epimastigotes (Figure 1). In other figure legends the authors mention the pulsation cycle. Were hypoosmotic conditions used for the other figures too or those were changes observed in different wild type or mutant cells under isosmotic condition? How the osmolarity was calculated, was an osmometer used or is there a reference for the method?
2. Introduction: Page 3, second paragraph: use "early divergent" rather than "primitive". Page 4, second paragraph: change the sentence "the overall morphology and membrane biochemistry ...similar to other trypanosomatids" Only the membrane biochemistry of *T. cruzi* CVC has been apparently studied among trypanosomatids. If not, provide references.
3. Page 7: indicate in Methods how the PDEC2 antibody was obtained or provide a reference.
4. Page 9: Figures 5H-I are mentioned in the text but not shown in the Figure.
5. The Discussion mentions fusion of the spongiome to the central vacuole (bladder). However, what is the evidence that the spongiome is not always in communication to the bladder? The system has been compared to a glove where the fingers are the spongiome and the bladder is the hand. When the glove is inflated, the fingers are used to increase the bladder surface and less spongiome is observed.
6. Page 13: The V-H⁺-ATPase has been proposed as the pump resulting in IMPs in *Paramecium*.
7. Page 15: provide references for the overexpression of PDEC2 and Vps34.
8. The Discussion could be shortened considerably.

Reviewer #2 (Comments to the Authors (Required)):

This manuscript by Kildare Miranda and colleagues presents a detailed structural analysis of the Contractile Vacuole Complex (CVC) of the kinetoplastid parasite *Trypanosoma cruzi*. The CVC is an organelle present in various lower eukaryotes that plays an important role in Regulatory Volume Decrease (RVD) that occurs when parasites are exposed to osmotic stress, e.g., hypotonic conditions. *T. cruzi* undergo dramatic changes in osmotic environment during their life cycle, living in highly hypertonic conditions in the insect vector hindgut and being delivered to environments of lower tonicity in the mammalian host, and the CVC provides one important mechanism for adaptation and survival. To re-establish homeostasis, the parasites use this organelle to extract water from the cytosol and export it from the cell through the flagellar pocket located at the base of the flagellum. While this organelle has been recognized for some years, this paper contributes significantly to understanding its structure and function in greater detail and using more sophisticated approaches than previous work. In particular, the authors utilize cutting edge electron microscopy, specifically high pressure freezing and freeze-substitution, cryo-EM, and tomography, and two mutants with altered CVC activity to map out the interactions of various CVC components and to propose a temporal process whereby the CVC collects water and expels it from the parasite. This work will be of interest to parasitologists and to workers on other single cell eukaryotes that possess structurally related CVCs as a way of maintaining essential osmotic balance under varying environmental conditions.

To study the function of the CVC, parasites were exposed to both isosmotic and hypoosmotic conditions, demonstrating that the Central Vacuole (CV) of the CVC is smaller in volume and elliptical (which they designate systole) under isosmotic conditions and larger and circular (diastole) under hypoosmotic conditions, the latter representing filling of the CV with water. The spongiome (Sp), a network of tubules that connect to the CV, extend into the cytosol and increase in volume under hypoosmotic conditions, suggesting that they collect water from the cytosol and deliver it, via visible Sp-CV connections, to the CV when

osmolarity drops.

A mutant that overexpresses a cAMP phosphodiesterase, TcrPDEC2, has reduced RVD, whereas a mutant that overexpresses a PI3 kinase, TcVps34, has hyper efficient RVD. These mutants were also imaged by high pressure freezing and freeze substitution (to maintain accurate subcellular morphology) to support a model of CVC activity. The hypo-active PDEC2 mutant showed small, disconnected Sp tubules and smaller CV compared to WT parasites, whereas the hyper-active Vps34 overexpressor showed larger Sp tubules that were strongly connected to the larger CV. These observations support the temporal model for CVC activity outlined in the paragraph above.

Additional significant observations focus on the adhesion plaque, a dense structure that connects the CV to the flagellar pocket (FP) to which water is delivered for expulsion from the cell. 3D tomographic imaging of this structure shows, for the first time, that it contains a fibrillar structure that connects the two organellar membranes. An aperture in the adhesion plaque can be observed in some parasites and suggests that this connection allows expulsion of water from the CV into the FP. While one obviously cannot observe directly the temporal changes proposed to occur for the CVC using EM, the data from isosmotic versus hypoosmotic conditions and from the two overexpressing mutants makes a consistent story that supports this mechanistic model and that adds significantly to our understanding of this regulatory organelle.

While the paper presents elegant data and a highly plausible model, it could be improved by some modifications in the presentation. In particular, adding more details in figure legends and in some places in the text will help readers, especially those who are not EM experts, to know how the experiments were performed and increase the transparency of the experimental approach.

Specific Suggestions.

1. Figure 5 shows 3D models of the CVC in systole and diastole in WT, PDEC2, and Vps34 mutants. Were these images generated from single cells of each line under the two conditions (please state so in the figure legend)? If so, has the modeling been repeated for multiple cells to confirm that the structural changes observed are reproducible and consistent?
2. Have the results in Fig. 4 for the 3 parasite lines been replicated on more than the 3 images shown in this figure and are they representative of the results from multiple cells?
3. Similarly, in Tables 2 and 3, are the numbers displayed from single images of CVCs, and if so, have any replicates been examined to test how representative the data are?
4. In Fig. 5I, a bar should connect the two rectangles that show statistically significant differences.
5. For each figure displaying EM images, the figure legend should more clearly state the method of sample preparation and data collection that was employed. In Fig. 1 was high pressure freezing used before freeze-substitution (presumably it was). In Fig. 7 A-I, are the images from tomograms? Similar descriptions should accompany each of the other figures.
6. Some markings are either absent from the figures or are not displayed clearly enough to see. In Fig. 5 G,H, where is the asterisk that is indicated in the figure legend? In Fig. 10F, I do not see the arrows indicated in the legend. Also, where is panel H for this figure?
7. On page 4, the 3rd paragraph, it states that cAMP activates protein kinase A. However, in kinetoplastids, as shown conclusively for *T. brucei*, PKA is not activated by cAMP but by purine nucleosides.
8. In the Materials and Methods section, there are multiple edits that should be made, as indicated below.

Pg. 16. 'proteins were resolved'

Was TcVps34 tagged with HA? This should be indicated

Pg. 17. Were the ET ribbons serial sections from freeze-substituted resin embedded material?

Pg. 18, bottom. 'epimastigote cells'.

Pg. 19. 1st paragraph. 'Quantification of the volume and continuous (?) area'....

'The complete volume of CVCs was acquired' (change word order)

'totaling 30 cells analyzed'

2nd paragraph. 'Nine to 15 models were obtained'...

'the high variability found in this structure'

3rd paragraph. 'virtual sections of peripheral and central regions of plaques from a total of 40 filaments'...

5th paragraph. 'the filamentous network observed inside the adhesion plaque'...

Reviewer #1

- 1. The Materials and Methods section indicates that hypoosmotic conditions were used only with epimastigotes (Figure 1). In other figure legends the authors mention the pulsation cycle. Were hypoosmotic conditions used for the other figures too or those were changes observed in different wild type or mutant cells under isosmotic condition? How the osmolarity was calculated, was an osmometer used or is there a reference for the method?**

Answer: The changes observed in wild-type and mutant cells in figures other than Figures 1 and 2, which refer only to wild-type cells, were under isosmotic conditions. Since *T. cruzi* does not exhibit a synchronized pulsation cycle, we can observe cells in different phases of the CVC within the same sample. We added a sentence in the discussion to clarify this on page 12.

We added information about the osmometer used to measure the osmolarity in the 'Hypoosmotic Treatment' section of the Material and Methods, as well as a reference to the study that served as the basis for these experiments.

- 2. Introduction: Page 3, second paragraph: use "early divergent" rather than "primitive". Page 4, second paragraph: change the sentence "the overall morphology and membrane biochemistry ...similar to other trypanosomatids" Only the membrane biochemistry of T. cruzi CVC has been apparently studied among trypanosomatids. If not, provide references.**

Answer: Modified accordingly on Page 3. The mentioned sentence on page 4 was excluded.

- 3. Page 7: indicate in Methods how the PDEC2 antibody was obtained or provide a reference.**

Answer: Reference added to "methods" – page 16.

- 4. Page 9: Figures 5H-I are mentioned in the text but not shown in the Figure.**

Answer: Figures 5H and 5I were shown on page 27 of the original manuscript. They refer to the volumes measured in WT and Vps34 cells, respectively.

5. The Discussion mentions fusion of the spongiome to the central vacuole (bladder). However, what is the evidence that the spongiome is not always in communication to the bladder? The system has been compared to a glove where the fingers are the spongiome and the bladder is the hand. When the glove is inflated, the fingers are used to increase the bladder surface and less spongiome is observed.

Answer: Part of the spongiome is always in connection with the bladder. Nevertheless, some of the tubules and vesicles are frequently seen disconnected. This was observed in the dozens of serial tomograms of various cells. In the ms, we depicted a few (representative images in Fig. 5) and extracted quantitative data from the models (tables 2 and 3). Briefly, our data points to fewer connections with the bladder in the first phase of systole, which progressively increase in number through fusions, while the spongiome elements (previously disconnected from the bladder) increase in volume by capturing water. As the fusions occur, the number of free vesicles and tubules in the spongiome decreases, and the bladder size and volume increase. This is schematically shown in Figure 2E. Reconstructions shown mainly in Figure 5, as well as Tables 2 and 3, represent what was thoroughly observed in ETs datasets.

6. Page 13: The V-H+-ATPase has been proposed as the pump resulting in IMPs in Paramecium.

Answer: Thanks. This information and its respective reference were included accordingly.

7. Page 15: provide references for the overexpression of PDEC2 and Vps34.

Answer: References added accordingly.

Reviewer #2:

- 1. Figure 5 shows 3D models of the CVC in systole and diastole in WT, PDEC2, and Vps34 mutants. Were these images generated from single cells of each line under the two conditions (please state so in the figure legend)? If so, has the modeling been repeated for multiple cells to confirm that the structural changes observed are reproducible and consistent?**

Answer: Yes, the 3D models shown in Fig 5 were obtained from single cells of each line in both systole and diastole conditions. They are, nevertheless, representative of many other tomograms obtained from multiple cells. This information was included in the figure legend, as suggested. As to the quantifications shown in Figures 5G-I, they were carried out on serial tomograms obtained from **five** randomly selected cells per each cell line (**total of 15**) in both systole and diastole stages (**total of 30 serial tomograms**), as stated in the legend of Figure 5 and in the "Morphometric and Statistical Analyses" section of Materials and Methods. The observed morphology was consistent across all samples both in thin sections and in datasets of different serial tomograms, including those (five) selected to generate the models and extract quantitative information for each cell line in each condition.

- 2. Have the results in Fig. 4 for the 3 parasite lines been replicated on more than the 3 images shown in this figure and are they representative of the results from multiple cells?**

Answer: Yes, the phenotypes shown in Fig 4 were observed in many, many cells on thin sections and tomograms obtained by transmission ET or FIB-SEM tomography. The usual observations comprise: i) the location of the CVC adjacent to the flagellar pocket in WT cells and both mutant lines; ii) the presence of the adhesion plaque also in the mutants; and iii) the different sizes of the CVC among the groups (the mutants and the wild type cells). All subsequent figures and data in the manuscript, obtained from different cells and experiments, confirmed these initial observations. Figures 7A and G, 8A and G, and 10F and G showed different mutant cells with the CVC adjacent to the flagellar pocket and the presence of the adhesion plaque. Figure 5, as discussed in the answer to question 1, displayed 3D models from other cells analyzed both qualitatively and quantitatively, showing the smaller size of the CVC in TcrPDEC2 OE and the larger CVC in TcVps34 OE, confirming our observations from the ultrathin sections. A sentence was added to the legend for better clarity.

- 3. Similarly, in Tables 2 and 3, are the numbers displayed from single images of CVCs, and if so, have any replicates been examined to test how representative the data are?**

Answer: No. The data displayed in Tables 2 and 3 refer to the number of the fragments (classified by size) from **five 3D models** obtained from **each cell line** (total of 15) in each condition - systole and diastole – reaching a total of **30 serial tomograms**. The data corroborate our observations in different datasets, whose representative images are shown in Figure 5. Quantitative information, however, was extracted from 30 models. For clarity, we added the information to the legends to the tables.

- 4. In Fig. 5I, a bar should connect the two rectangles that show statistically significant differences.**

A bar was not included in Fig. 5I because the significant difference is related to the rectangle representing the total volume of the WT, displayed in the Fig. 5H. This has been clarified in the legend.

- 5. For each figure displaying EM images, the figure legend should more clearly state the method of sample preparation and data collection that was employed. In Fig. 1 was high pressure freezing used before freeze-substitution (presumably it was). In Fig. 7 A-I, are the images from tomograms? Similar descriptions should accompany each of the other figures.**

Answer: Thank you for pointing this out. The methods used for sample preparation and analysis were added to the figure legends accordingly.

- 6. Some markings are either absent from the figures or are not displayed clearly enough to see. In Fig. 5 G,H, where is the asterisk that is indicated in the figure legend? In Fig. 10F, I do not see the arrows indicated in the legend. Also, where is panel H for this figure?**

Answer: In Figure 5, the asterisk is shown only in 5I. This has been explained above and the legend modified. Figure 10 was modified to include the missing arrows and the corresponding changes have been noted in the legend. We also changed Figure 9 by increasing the thickness of the inset lines. Regarding panel H, we apologize for the mistake. This was a leftover from another version of the figure.

- 7. On page 4, the 3rd paragraph, it states that cAMP activates protein kinase A. However, in kinetoplastids, as shown conclusively for T. brucei, PKA is not activated by cAMP but by purine nucleosides.**

Thank you for clarifying this point. We modified the paragraph to reflect the current model of the osmoregulation pathway in *T. cruzi*, including a PKA-independent pathway while still highlighting the role of cAMP in osmoregulation by inducing the fusion of acidocalcisomes with the CVC, as suggested by Chiurillo et al. (2023) and Docampo (2024). Conversely, in *T. brucei*, cAMP is involved in different cellular processes, and PKA activation is cAMP-independent, based on purine nucleosides, as pointed out by the reviewer and demonstrated by Bachmaier et al. (2019). For better clarity, we excluded the mention of PKA in the osmoregulation pathway.

- 8. In the Materials and Methods section, there are multiple edits that should be made, as indicated below.**

Pg. 16. 'proteins were resolved'

Was TcVps34 tagged with HA? This should be indicated

Pg. 17. Were the ET ribbons serial sections from freeze-substituted resin embedded material?

Pg. 18, bottom. 'epimastigote cells'.

Pg. 19. 1st paragraph. 'Quantification of the volume and continuous (?) area'....

'The complete volume of CVCs was acquired' (change word order)

'totaling 30 cells analyzed'

2nd paragraph. 'Nine to 15 models were obtained'...

'the high variability found in this structure'

3rd paragraph. 'virtual sections of peripheral and central regions of plaques from a total of 40 filaments'...

5th paragraph. 'the filamentous network observed inside the adhesion plaque'...

Answer: All corrections were done.

July 15, 2024

RE: Life Science Alliance Manuscript #LSA-2024-02826-TR

Dr. Kildare Miranda
Federal University of Rio de Janeiro
Biophysics Institute
Av. Carlos Chagas Filho, 373, Bl C
Cidade Universitária
Rio de Janeiro, Rio de Janeiro 21941902
Brazil

Dear Dr. Miranda,

Thank you for submitting your revised manuscript entitled "Quantitative assessment of the nanoanatomy of the contractile vacuole complex in *Trypanosoma cruzi*". We would be happy to publish your paper in Life Science Alliance pending final revisions necessary to meet our formatting guidelines.

- please be sure that the authorship listing and order is correct
- please add a Summary Blurb in our system
- please add the Twitter handle of your host institute/organization as well as your own or/and one of the authors in our system
- please add a Conflict of Interest statement to your main manuscript text
- please label the Figures in the system as Figure 1, Figure S1, Figure 2, Figure S2, etc.
- please use the [10 author names, et al.] format in your references (i.e. limit the author names to the first 10)
- please place main manuscript tables at the end of manuscript file after Figure legends and Supplementary Figure Legends and Video legends
- supplementary tables should be uploaded separately, and not included in main manuscript file (either in .docx or excel file format)
- please label the videos in the system as Video S1, Video S2 etc.
- please include labels A-B in the caption of Figure S2, as they are present in the figure itself
- Fig S4A is cited in manuscript text, while there is no Figure S4. Please rectify this discrepancy
- please add a callout for Figures 5D, 6A-D, 8A-I, S1A-B, S2A and S3A-E to your main manuscript text

A. FINAL FILES:

-- Summary blurb (enter in submission system): A short text summarizing in a single sentence the study (max. 200 characters including spaces). This text is used in conjunction with the titles of papers, hence should be informative and complementary to the title. It should describe the context and significance of the findings for a general readership; it should be written in the

present tense and refer to the work in the third person. Author names should not be mentioned.

B. MANUSCRIPT ORGANIZATION AND FORMATTING:

Sincerely,

July 18, 2024

RE: Life Science Alliance Manuscript #LSA-2024-02826-TRR

Dr. Kildare Miranda
Federal University of Rio de Janeiro
Biophysics Institute
Av. Carlos Chagas Filho, 373, BI C
Cidade Universitária
Rio de Janeiro, Rio de Janeiro 21941902
Brazil

Dear Dr. Miranda,

Thank you for submitting your Resource entitled "Quantitative assessment of the nanoanatomy of the contractile vacuole complex in *Trypanosoma cruzi*". It is a pleasure to let you know that your manuscript is now accepted for publication in Life Science Alliance. Congratulations on this interesting work.

DISTRIBUTION OF MATERIALS:

Again, congratulations on a very nice paper. I hope you found the review process to be constructive and are pleased with how the manuscript was handled editorially. We look forward to future exciting submissions from your lab.

Sincerely,
